# Ocean Carbon Uptake Under Aggressive Emission Mitigation

Sean Ridge[1] and Galen A. McKinley[1]

[1]Columbia University and Lamont Doherty Earth Observatory, New York, United States

**Correspondence:** (Galen A. McKinley, mckinley@ldeo.columbia.edu)

**Abstract.** Nearly every nation has signed the UNFCC Paris Agreement, committing to mitigate anthropogenic carbon emissions so as to limit the global mean temperature increase above pre-industrial to well below $2°C$, and ideally to no more than $1.5°C$. A consequence of emission mitigation that has received limited attention is a reduced efficiency of the ocean carbon sink. Historically, the roughly exponential increase of atmospheric $CO_2$ has resulted in a proportional increase in anthropogenic carbon uptake by the ocean. We define growth of the ocean carbon sink exactly proportional to the atmospheric growth rate to be 100% efficient. Using a model hierarchy consisting of a common reduced-form ocean carbon cycle model and the Community Earth System Model (CESM), we assess the mechanisms of future change in the efficiency of the ocean carbon sink under three emission scenarios: aggressive mitigation ($1.5°C$), intermediate mitigation (RCP4.5), and high emissions (RCP8.5). The reduced-form ocean carbon cycle model is tuned to emulate the global-mean behavior of the CESM, and then allows for mechanistic decomposition. With intermediate or no mitigation (RCP4.5, RCP8.5), changes in efficiency through 2080 are almost entirely the result of future reductions in the carbonate buffer capacity of the ocean. Under the $1.5°C$ scenario, the dominant driver of efficiency decline is the ocean's reduced ability to transport anthropogenic carbon from surface to depth. As the global-mean upper-ocean gradient of anthropogenic carbon reverses sign, carbon can be re-entrained in surface waters where it slows further removal from the atmosphere. Reducing uncertainty in ocean circulation is critical to better understanding the transport of anthropogenic carbon from surface to depth, and to improving quantification of its role in the future ocean carbon sink.

## 1 Introduction

The ocean has absorbed excess carbon equivalent to 39% of the $CO_2$ from industrial era fossil fuel combustion and cement production (Friedlingstein et al., 2019). The rest of the $CO_2$ remains in the atmosphere where it acts as the primary driver of climate change. At the global scale, the partial pressure of $CO_2$ in the atmosphere ($pCO_2^{atm}$) is greater than the partial pressure of $CO_2$ in the surface ocean ($pCO_2^{ocn}$), thus there is a net ocean sink. The difference in partial pressures has grown over time, therefore ocean uptake of atmospheric $CO_2$ has increased over the industrial era (Khatiwala et al., 2009; DeVries, 2014). The carbon that has been added to the ocean and atmosphere as the result of anthropogenic $CO_2$ emissions is referred to as anthropogenic carbon, $C_{ant}$.

The rate of ocean anthropogenic carbon uptake is further controlled by carbon chemistry in seawater and physical removal of anthropogenic carbon from the surface ocean into the ocean interior (Graven et al., 2012). Various processes set the rate

of transport from surface to depth of anthropogenic carbon (Bopp et al., 2015; Gnanadesikan et al., 2015; Iudicone et al., 2016). Advection and watermass transformation dominates regional patterns of anthropogenic carbon fluxes into (reemergence) and out of (subduction) the seasonal mixed layer (Bopp et al., 2015; Iudicone et al., 2016; Toyama et al., 2017). However, large positive and negative signs of these fluxes mostly cancel when globally integrated (Bopp et al., 2015), and thus can be conceptualized as a diffusive process in a vertical column (Section 2.4). By parameterizing the ocean's global-mean removal of carbon to depth as a constant process, models based on an impulse response function (IRF) can replicate ocean anthropogenic carbon uptake that is quantitatively consistent with the uptake of complex models and observations (Oeschger et al., 1975; Joos et al., 1996).

The efficiency of land and ocean sinks may be described by the $CO_2$ sink rate ($k_S$; Raupach et al. (2014)), which is the combined ocean-land $CO_2$ uptake per unit atmospheric $CO_2$ above preindustrial levels ($C_{ant}^{ATM}$, PgC):

$$k_S(t) = \frac{F_{ant}^L(t) + F_{ant}^M(t)}{C_{ant}^{ATM}(t)} \tag{1}$$

Where $F_{ant}^L$ (Pg C yr$^{-1}$) is the anthropogenic land sink and $F_{ant}^M$ (Pg C yr$^{-1}$) is the anthropogenic ocean sink. Observations of $k_S$ from 1959-2012 indicate a robust declining trend, and thus the rate of increase in the sinks has been slower than the accumulation of carbon in the atmosphere (Canadell et al., 2007; Raupach et al., 2014). Raupach et al. (2014) illustrate that the observed declining $k_S$ is attributable to this slower-than-exponential $CO_2$ emissions growth ($\sim$35% of the trend), a decline in major volcanic eruptions, which cause brief periods of global cooling ($\sim$25%), response of the natural sinks to a warming climate ($\sim$20%), and nonlinear responses to increasing atmospheric $CO_2$ (mostly attributable to ocean chemistry; $\sim$20%). The contribution of ocean anthropogenic carbon uptake to $k_S$ is $k_M$:

$$k_M(t) = \frac{F_{ant}^M(t)}{C_{ant}^{ATM}(t)} \tag{2}$$

If there is exponentially increasing $pCO_2^{atm}$, and constant gas solubility, air-sea transfer coefficient, and carbonate buffer capacity, theory indicates that $k_M$ will remain constant (Raupach et al., 2014). Because these conditions do approximately describe historical conditions, constant proportionality for ocean anthropogenic carbon uptake has been used as a null hypothesis in studies of the drivers of historical regional and global scale changes in the ocean carbon cycle (Lovenduski et al., 2008; Gruber et al., 2019). Here we refer to this constant proportionality (i.e. $k_M = constant$) as the "historical scaling". The term often used is "the transient steady state assumption" (Gammon et al., 1982; Tanhua et al., 2007; Lovenduski et al., 2008; Gruber et al., 2019). We choose "historical scaling" to clarify that this null hypothesis was appropriate for the last several decades of the 20th century, and to allow for emphasis on the fact that this assumption should be increasingly less accurate going forward.

Slowing of the emissions growth rate, and thus the $pCO_2^{atm}$ growth rate, reduces the efficiency of $k_M$ (Raupach et al., 2014; McKinley et al., 2020). A central motivation for this work is the fact that, in the future, a reduced $pCO_2^{atm}$ growth rate is inevitable, due either to climate policy (Hausfather and Peters, 2020) or to the eventual exhaustion of fossil fuel reservoirs.

In addition to $k_M$ efficiency changes due to slowing $pCO_2^{atm}$ growth rate, there will also be impacts on $k_M$ from carbon cycle feedbacks (Friedlingstein et al., 2013; Raupach et al., 2014). Past studies have separated carbon cycle feedbacks into $CO_2$ concentration effects and climate driven effects (Friedlingstein et al., 2013; Arora et al., 2013). Climate driven effects

stem from the warming of the surface ocean, which reduces gas solubility and slows the ocean circulation, thus reducing the efficiency of ocean uptake (Friedlingstein et al., 2013). The $CO_2$ concentration effect in the ocean has typically been thought of as the net result of two effects: increased flux driven by increasing $pCO_2^{atm}$ and reduced flux due to declining buffer capacity. The buffering capacity of the ocean refers to the transfer of absorbed $CO_2$ via chemical reactions into chemical species that do not exchange with the atmosphere. As more $CO_2$ is added to the ocean, buffer capacity decreases (Fassbender et al., 2017). When buffer capacity is reduced, more of the $CO_2$ remains in a form that can exchange with the atmosphere, and thus the efficiency of carbon uptake declines. Schwinger and Tjiputra (2018) illustrate that for scenarios of emission mitigation, there is also an important additional component to the $CO_2$ concentration feedback. Because the ocean only slowly transports $CO_2$ from surface to depth, when emissions are mitigated, the elevated $CO_2$ concentration of the upper ocean acts to slow additional carbon uptake. We explore this feedback under more realistic forcing scenarios in this study.

This work expands upon previous work that has quantified future change in ocean anthropogenic carbon uptake. We separately account for changes due to buffering, due to the impact of warming on solubility and inorganic carbonate chemistry, and due to the future trajectory of $pCO_2^{atm}$. By residual, we can then estimate the degree to which carbon already held in the upper ocean will slow the sink. We compare future scenarios with high emissions (RCP8.5), intermediate mitigation (RCP4.5; Meinshausen et al. (2011)) and an aggressive mitigation scenario where the 1.5°C target is met (1.5°C; Sanderson et al. (2017)) using ensemble results from an Earth System Model (ESM). We use a reduced-form ocean carbon cycle model to emulate the ESM for each scenario, and with it, diagnose the mechanisms of ocean carbon sink efficiency decline in the future projections. We determine for the three $pCO_2^{atm}$ scenarios how reduced buffering, warming impacts on carbon solubility and inorganic carbonate chemistry, and a steady circulation interacting with a changing surface to depth gradient of anthropogenic carbon should impact ocean anthropogenic carbon uptake through 2080.

## 2   Methods

### 2.1   Efficiency Metric and Historical Scaling

Efficiency, $\eta$, is $k_M$ (Equation 2) referenced to the year 1990, and expressed as a percentage:

$$\eta(t) = \frac{k_M(t)}{k_M(1990)} \times 100 \tag{3}$$

Referencing $k_M$ to 1990 values maximizes the time ocean anthropogenic carbon uptake is at 100% efficiency during the historical period, 1920-2006 (Figure S1). The historical scaling for ocean anthropogenic carbon air-sea flux ($F_{ant}$) is closely related to $k_M$:

$$\overset{*}{F}_{ant}(t) = k_M(1990)C_{ant}^{ATM}(t) = F_{ant}(1990)\frac{C_{ant}^{ATM}(t)}{C_{ant}^{ATM}(1990)} \tag{4}$$

The overset "*" notation indicates the variable that has been extrapolated with the historical scaling. Here, $F_{ant}(1990)$ is diagnosed from the CESM simulations. Following from Equation 3, ocean carbon sink efficiency ($\eta$) is related to the historical

scaling:

$$\eta(t) = \frac{F_{ant}(t)}{\overset{*}{F}_{ant}(t)} \times 100 \tag{5}$$

Under the approximately exponential $pCO_2^{atm}$ increase of the historical period, $k_M$ is relatively constant, thus $F_{ant}(t) \approx \overset{*}{F}_{ant}(t)$ and historical period efficiency is $\sim 100\%$. Because it is approximately equal to $F_{ant}$, $\overset{*}{F}_{ant}$ has been used to estimate historical $F_{ant}(t)$ (Lovenduski et al., 2008). In the future, as $k_M$ declines from 1990 values, $F_{ant}$ will be less than $\overset{*}{F}_{ant}(t)$, i.e. efficiency will decline. In this study, $\overset{*}{F}_{ant}(t)$, extrapolated into the future with projected $pCO_2^{atm}$, is taken as a useful reference point against which to compare projected future ocean anthropogenic carbon uptake.

We also use the historical scaling as a baseline for determining anthropogenic carbon concentration ($C_{ant}(x,y,z,t)$) changes in the interior (Tanhua et al., 2007; Gruber et al., 2019) in CESM:

$$\overset{*}{C}_{ant}(x,y,z,t) = C_{ant}(x,y,z,1990)\frac{C_{ant}^{ATM}(t)}{C_{ant}^{ATM}(1990)} \tag{6}$$

We use reference anthropogenic carbon concentrations ($C_{ant}(x,y,z,1990)$) from the CESM simulations. The $C_{ant}(x,y,z,t)$ historical scaling exists because the exponential signal of atmospheric $CO_2$ increase is transmitted by the air-sea flux of anthropogenic carbon to surface ocean mixed layer anthropogenic carbon concentration ($C_{ant}^{ML}$), and then ocean circulation passes the exponential signal into the interior. $C_{ant}^{ML}$ is closely related to the time integral of the air-sea flux of anthropogenic carbon (Section 2.3). Because the integral of an exponential is also an exponential, $C_{ant}^{ML}$ has also increased exponentially. From the surface, the exponential atmospheric signal is propagated to deeper layers by the ocean circulation.

With this work, we study the three processes that will cause the ocean carbon sink to diverge from its historical scaling in the coming decades, through 2080. First, the linear relationship between increasing $C_{ant}^{ML}$ and $pCO_2^{ocn}$ will end due to the decreasing buffer capacity for $CO_2$. Second, warming of the surface ocean will cause reduced $CO_2$ solubility and modify inorganic carbonate chemistry. Third, if emissions are mitigated, $C_{ant}^{ML}$ will fall, but slightly deeper waters will still contain the higher $C_{ant}$ concentrations set by the atmospheric $CO_2$ of decades prior. There will thus be a "back-pressure" on $C_{ant}^{ML}$ coming from near-surface water that reemerge at the surface (Bopp et al., 2015; Iudicone et al., 2016). Our assessment of this back-pressure effect does not require change in the ocean circulation, as our decomposition assumes a circulation to be constant. Instead, this back pressure can be explained by the relatively slow rate at which the ocean redistributes $C_{ant}$ from surface to depth.

## 2.2 Ocean Component of the Earth System Model

We use the Community Earth System Model 1 (Hurrell et al., 2013) for our analysis of the three-dimensional ocean carbon sink. CESM's ocean component model, POP2, provides the three-dimensional, time-evolving estimates of the ocean carbon cycle (Long et al., 2013). POP2 output is from publicly available CESM climate simulations provided by the National Center for Atmospheric Research (NCAR). POP2 features 60 vertical levels and a nominal $1° \times 1°$ horizontal resolution. Surface boundary layer physics are parameterized using the K-Profile Parameterization (KPP) of Large et al. (1994). Unresolved advection

by eddies is parameterized with the Gent-McWilliams parameterization (Gent and Mcwilliams, 1990). Isopycnal mixing is parameterized with the Redi (1982) diffusion operator. The biogeochemical output comes from the embedded Biogeochemical Elemental Cycle (BEC) model (Moore et al., 2004). Anthropogenic carbon concentration is calculated in the model as the difference between natural carbon, a tracer that experiences a fixed preindustrial $pCO_2^{atm}$, and contemporary carbon, a tracer that experiences time evolving $pCO_2^{atm}$.

Following a long preindustrial spin-up, all simulations used here are forced for the historical period (1850-2005) with observations of $pCO_2^{atm}$. For 2006-2080, forcing is $pCO_2^{atm}$ from the Representative Concentration Pathways (RCPs) or a 1.5°C scenario (Sanderson et al., 2017). For the 1.5°C scenario, a concentration pathway was designed that limited warming the CESM to 1.5°C, for the purpose of investigating avoided climate impacts (Sanderson et al., 2017). This scenario features the same forcing as RCP8.5 until 2017, except for $CO_2$. Unfortunately, the projected $CO_2$ forcing was not smoothly joined to the historical $CO_2$ forcing, creating a period of anomalously low anthropogenic carbon flux from 2006 to 2017 (Figure S2). To avoid this unrealistic feature in our main figures, we plot the 1.5°C scenario only after 2017.

Multiple CESM simulations are run with the same $pCO_2^{atm}$ forcing to generate single model ensembles for each scenario. The ensemble approach allows for separation of internal variability from the forced signals, with the latter being the focus of this study. NCAR has run multiple ensembles with different forcings including CESM Large Ensemble (40 members, RCP8.5; Kay et al. (2015)), CESM Medium Ensemble (15 members, RCP4.5), and the CESM Low-Warming Ensemble (10 members, 1.5°C; Sanderson et al. (2017)). Individual ensemble members are branched off at 1920 (Kay et al., 2015). Ocean biogeochemistry output is limited to 9 members for the medium ensemble and the 3 for the low warming ensemble. To ensure a comparable number of ensemble members across our analysis, we use only 9 ensemble members for RCP8.5.

In coupled climate models, historical climate variability of the carbon sink is not expected to match observations because the phasing of ENSO or other internal climate variability is different in each ensemble member. Averaging across an ensemble removes the imprint of internal variability to reveal the response to external forcing (Kay et al., 2015). With only a single coupled climate simulation, decadal means would typically be used to smooth internal climate variability. However, since we are using an ensemble mean in which this variability has already been removed, the single years that we plot provide a snapshot of the climate response to external forcing. In this study, these CESM ensembles are used for all maps and sections. As explained below, we tune the reduced-form model to replicate the CESM's air-sea $CO_2$ flux ($F_{ant}$) under each scenario, and then use the reduced-form model to decompose the mechanisms for future change in sink efficiency.

## 2.3 Impulse Response Function Model for the Ocean Carbon Sink

We employ an established reduced form ocean carbon cycle model based on an impulse response function (IRF). This model has been used for decades to emulate ocean carbon uptake simulated by complex ESMs (Joos et al., 1996; Raupach et al., 2014), and is also used for all RCP scenarios to convert projected emissions to $CO_2$ concentrations (Meinshausen et al., 2011).

Impulse response functions characterize the dynamic system response to small perturbations around a steady state, with the full response being the sum of infinite discrete pulses. For the global-mean ocean carbon cycle, a pulse of anthropogenic carbon added to the surface ocean by air-sea exchange and the impulse response function determines the timescale with which that

pulse moves to deeper ocean layers. Surface ocean anthropogenic carbon content is solved as the convolution integral of the air-sea flux ($F_{ant}$, the impulse) and the lifetime of that anthropogenic carbon pulse ($r(t)$, the impulse response function):

$$C_{ant}^{ML}(t) = \frac{c}{h \cdot A_{oc}} \int_{t_i}^{t} F_{ant}(u) r(t-u) du \qquad (7)$$

The air-sea flux of anthropogenic carbon is dependent on the air-sea partial pressure gradient (ppm) and the gas exchange coefficient ($k_g$, $\text{yr}^{-1}$):

$$F_{ant} = k_g(pCO_2^{atm} - pCO_2^{ocn}) \qquad (8)$$

Where $pCO_2^{ocn}$ is the preindustrial $pCO_2^{ocn}$ ($pCO_2^{ocn,PI}$) plus an anthropogenic perturbation ($\delta pCO_2^{ocn}$), including effects of changing buffer capacity and temperature (Appendix A). Forcings are the historical and projected $pCO_2^{atm}$ that forced CESM, and historical and projected SST output by CESM.

The convolution integral in Equation 7 sets the concentration at time $t$ by calculating the fraction of previous pulses ($F_{ant}(u)$), that entered the ocean mixed layer at times prior ($t_i = 0$ to $t$). The effective mixed layer depth, $h$ is adjusted to tune the historical air-sea flux of anthropogenic carbon of the IRF model to emulate the historical ensemble-mean of CESM. CESM's historical flux is best replicated with $h = 51$m. We implement the impulse response function ($r(t)$) that was diagnosed by Joos et al. (1996, 2001) from the HILDA (HIgh Latitude-exchange/interior Diffusion-Advection) model. $r(t)$ is fixed in time, which is equivalent to assuming a constant circulation and background natural carbon cycle. There is a unit conversion factor ($c = 1.722$ $\mu$mol m$^3$ ppm$^{-1}$ kg$^{-1}$); and $A_{oc}$ is the ocean area (m$^2$). Directly diagnosing an ocean model's mixed layer impulse response function would require special simulations (Joos et al., 1996) that have not been performed for CESM. Instead, we show below that with the IRF from HILDA and $h$ as tuning parameter, we can emulate CESM behavior both historically and under these three future scenarios (Figure 2d). Thus, we can use this IRF to assist in separating the mechanisms of ocean carbon sink change that are occurring in the CESM projections. It is important to note that despite the ability of the IRF model to emulate CESM behavior for our period of study, this does not mean it should be expected to emulate CESM on longer timescales. Particularly under high emissions, greater ocean circulation and biogeochemical changes are expected beyond 2100 (Randerson et al., 2015).

### 2.4 Mechanistic Decomposition of the Air-Sea Flux

Considering anthropogenic perturbations on top of a background natural state in the surface ocean, the air-sea flux of anthropogenic carbon is a function of the $pCO_2$ in the atmosphere and ocean (Equation 8), and $pCO_2^{ocn}$ is a function of the anthropogenic carbon content ($C_{ant}$) and the temperature (T): $F_{ant}(pCO_2^{atm}, pCO_2^{ocn}(C_{ant}, T))$. Change in gas-exchange rates are assumed negligible, and because the biological pump is part of the background natural cycle, it is also assumed constant. The total derivative of the air-sea flux of anthropogenic carbon (Equation 8) can then be written in terms of its partial

derivatives:

$$\frac{dF_{ant}}{dt} = \underbrace{\overbrace{\frac{\partial pCO_2^{atm}}{\partial t}}^{atm.\ gr.\ rate} \frac{\partial F_{ant}}{\partial pCO_2^{atm}}}_{atmos.\ component} - \underbrace{\overbrace{\frac{\partial pCO_2^{ocn}}{\partial t}}^{ocn.\ gr.\ rate} \frac{\partial F_{ant}}{\partial pCO_2^{ocn}}}_{ocean\ component} \tag{9}$$

A positive $pCO_2^{atm}$ growth rate enhances $F_{ant}$, while positive $pCO_2^{ocn}$ growth acts to decrease $F_{ant}$. Since the $pCO_2^{atm}$ growth rate is prescribed, we further expand only the ocean component:

$$\frac{\partial pCO_2^{ocn}}{\partial t} = \frac{\partial C_{ant}}{\partial t} \frac{\partial pCO_2^{ocn}}{\partial C_{ant}} + \frac{\partial T}{\partial t} \frac{\partial pCO_2^{ocn}}{\partial T} \tag{10}$$

With the first term being the effect of the buffer factor and ocean circulation, and the second the sensitivity of $pCO_2^{ocn}$ to warming via solubility and inorganic carbonate chemistry. For the global-mean, the first term in Equation 10 can be further separated using:

$$\frac{\partial C_{ant}}{\partial t} = F_{ant} + K_z \frac{\partial C_{ant}}{\partial z} \tag{11}$$

Where $K_z$ is a vertical diffusivity representing the global-mean ocean circulation (Munk, 1966) acting on the vertical gradient of $C_{ant}$ in the ocean.

Substituting Equation 11 into Equation 10, we arrive at three terms controlling the evolution of $pCO_2^{ocn}$ :

$$\frac{\partial pCO_2^{ocn}}{\partial t} = \underbrace{F_{ant} \overbrace{\frac{\partial pCO_2^{ocn}}{\partial C_{ant}}}^{buffer\ factor}}_{impact\ of\ air-sea\ flux} + \underbrace{K_z \frac{\partial C_{ant}}{\partial z} \overbrace{\frac{\partial pCO_2^{ocn}}{\partial C_{ant}}}^{buffer\ factor}}_{impact\ of\ vertical\ C_{ant}\ transport} + \underbrace{\frac{\partial T}{\partial t} \overbrace{\frac{\partial pCO_2^{ocn}}{\partial T}}^{warm.\ sens.}}_{impact\ of\ warming} \tag{12}$$

On the right hand side, the first term is the impact of the air-sea flux on $pCO_2^{ocn}$, modulated by the buffer factor; the second the impact of ocean vertical transport, also modulated by the buffer factor, and the third the impact of warming on carbon chemistry. This conceptual decomposition is useful to understanding our experiments with the IRF model, explained in the following section.

## 2.5 Process Decomposition Using the Impulse Response Function Model

In CESM, $F_{ant}$, the vertical gradient of $C_{ant}$, the buffer factor, the circulation and the temperature are all evolving (Equation 12). Thus our emulation of CESM is the $C_{total}$ IRF experiment and it implicitly includes all these effects (Table 1; "All Effects (Control)"). We perform two sensitivity studies in which the temperature is held constant such that there are no impacts on carbon solubility and inorganic carbonate chemistry, $C_{nowarm}$ ("Constant Temperature"); and in which the buffer factor is held constant at a pre-industrial value and there is no warming, $C_{ccc}$ ("Constant Chemical Capacity"). The cumulative anthropogenic carbon uptake consistent with the historical scaling for each scenario is $\overset{*}{C}_{hs}$, calculated directly from the prescribed $pCO_2^{atm}$ (Table 1; Equation 4; "Historical Scaling"). Combining these experiments allows quantification of the three effects, $\Delta C_{warm}$, $\Delta C_{chem}$ and $\Delta C_{transp}$, that combine to make $C_{total}$ different from than the historical scaling ($\overset{*}{C}_{hs}$).

**Table 1.** Experiments with the IRF model, historical scaling, and the effects quantified by differencing.

| Experiment Name | Description | Symbol | Scenarios |
|---|---|---|---|
| All Effects (Control) | full chemistry, warming | $C_{total}$ | RCP8.5, RCP4.5, 1.5°C |
| Constant Temperature | constant temperature | $C_{nowarm}$ | RCP8.5, RCP4.5, 1.5°C |
| Constant Chemical Capacity | constant buffer factor | $C_{ccc}$ | RCP8.5, RCP4.5, 1.5°C |
| Historical Scaling[1] | constant efficiency | $\overset{*}{C}_{hs}$ | RCP8.5, RCP4.5, 1.5°C |

| Effect Name | Effect Symbol | Equation |
|---|---|---|
| Warming | $\Delta C_{warm}$ | $C_{total} - C_{nowarm}$ |
| Chemical Capacity | $\Delta C_{chem}$ | $C_{nowarm} - C_{ccc}$ |
| Vertical Transport of $C_{ant}$[2] | $\Delta C_{transp}$ | $C_{ccc} - \overset{*}{C}_{hs}$ |

1. $\overset{*}{C}_{hs}$ is calculated directly from $pCO_2^{atm}$ ($\overset{*}{C}_{hs} = \int \overset{*}{F}_{ant}\, dt$; Equation 4)

2. $\Delta C_{transp}$ is only defined when negative, i.e. when $\overset{*}{C}_{hs} > C_{ccc}$

Since the circulation is assumed constant, the change due to warming, $\Delta C_{warm}$ only accounts for the impact of warming on solubility and inorganic carbonate chemistry. Change in ocean chemical capacity, $\Delta C_{chem}$, is the change in anthropogenic carbon uptake in the IRF model simulation with full chemistry but no warming, $C_{nowarm}$, minus the change in anthropogenic carbon in the IRF model simulation with a constant buffer factor and no warming, $C_{ccc}$ (Table 1). Thus, $\Delta C_{chem}$ quantifies the impact of change in inorganic carbonate chemistry that occurs as additional $C_{ant}$ is absorbed.

When the historical scaling indicates greater carbon uptake by the ocean than the combined negative impacts of warming and chemistry, then the transport effect can be defined as the remaining difference:

$$C_{total} = \overset{*}{C}_{hs} + \Delta C_{warm} + \Delta C_{chem} + \Delta C_{transp} \text{ (when } \overset{*}{C}_{hs} > C_{ccc}) \tag{13}$$

     The impact of the vertical transport of $C_{ant}$ on the ocean sink is due to the sensitivity of the transport of anthropogenic carbon from surface to depth on the vertical profile of $C_{ant}$ (Equation 12). The physical circulation and background natural carbon

cycle are assumed fixed in the IRF model, consistent with the carbon cycle in CESM not illustrating significant sensitivity to such changes over 1920-2080 under RCP8.5 high-emission forcing (Randerson et al., 2015). However, particularly under aggressive mitigation, there is significant change in the vertical gradient of $C_{ant}$ on which this circulation will act, and thus the net effect of $\Delta C_{transport}$ will be to slow the ocean carbon sink.

     For much of the projected future under RCP8.5 and RCP4.5, $pCO_2^{atm}$ growth will, in fact, be greater than exponential

(Figure S3). Under these conditions, the estimate of carbon uptake by the historical scaling is less than the sum of the impact of warming and chemistry, and $\Delta C_{transp}$ cannot be sensibly defined. The upper bound on potential carbon uptake by the ocean in this case is $C_{ccc}$, our IRF experiment in which neither buffer capacity nor temperature change. Thus:

$$C_{total} = C_{ccc} + \Delta C_{warm} + \Delta C_{chem} \text{ (when } \overset{*}{C}_{hs} < C_{ccc}) \tag{14}$$

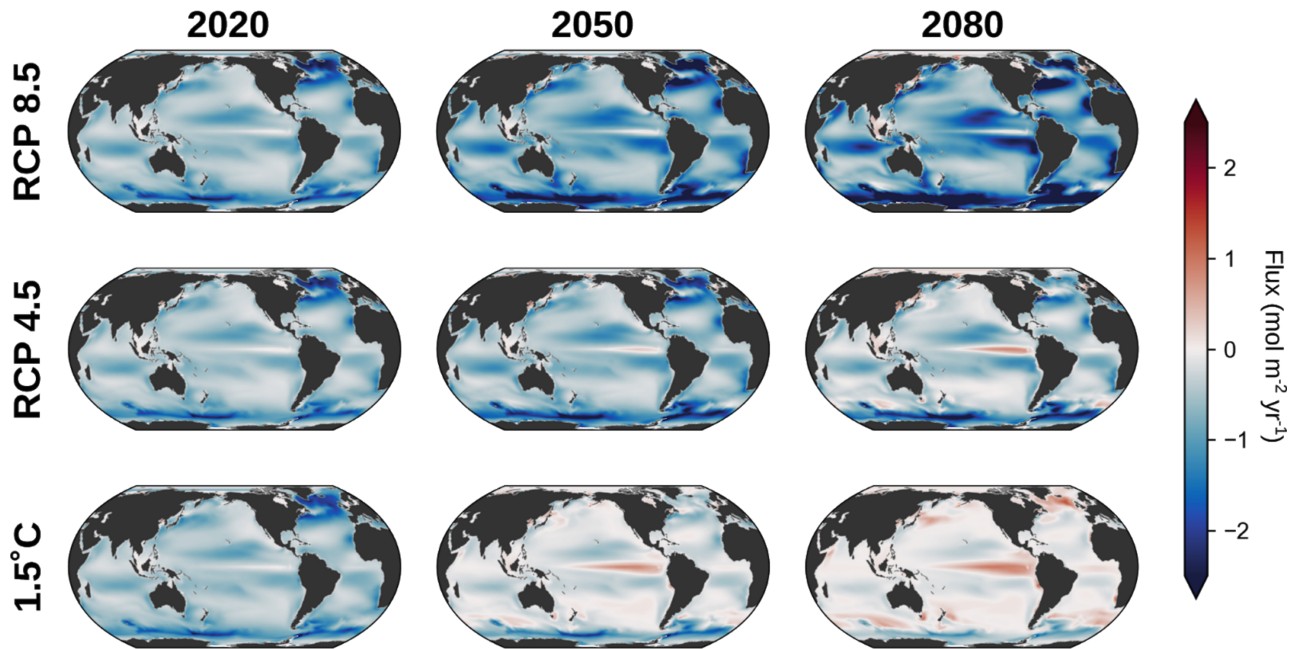

**Figure 1.** CESM ensemble-mean air-sea flux of anthropogenic carbon (mol $C_{ant}$ m$^{-2}$ yr$^{-1}$; positive = red = to the atmosphere). Each row is a scenario, and each column represents a year. Emission mitigation is greatest at the bottom of each column.

## 3 Results

### 3.1 Projected Spatial Patterns of Anthropogenic Air-Sea Carbon Flux

In CESM, the projected spatial distribution of the air-sea flux of anthropogenic carbon from 2020-2080 differs across the three future scenarios: 1.5°C, RCP4.5, RCP8.5.

In the 1.5°C scenario, the spatial pattern of the air-sea flux of anthropogenic carbon changes significantly from 2020-2080. While most of the ocean is a sink in 2020, in 2050 and 2080 there are large regions of anthropogenic carbon outgassing (Figure 1, bottom row). Most pronounced is the emergence of anthropogenic carbon outgassing in the equatorial Pacific. The outcrop region of Sub-Antarctic Mode Water (SAMW) at about 50°S also experiences outgassing by 2080. In 2020, the Kuroshio and subpolar North Atlantic are some of the most intense sinks of $C_{ant}$, but by 2080, these regions are sources. Contrastingly, Southern Ocean anthropogenic carbon uptake persists throughout the simulation.

In the RCP4.5 scenario, equatorial Pacific outgassing of anthropogenic carbon grows over time (Figure 1, middle row), but is less widespread and intense than in the 1.5°C scenario. The intensity of uptake flux decreases over time for the subpolar and mid-latitude Atlantic and Kuroshio region. Beyond the equatorial Pacific, the spatial pattern of the air-sea flux of anthropogenic carbon is similar to the RCP8.5 scenario, but the amplitude of uptake is reduced.

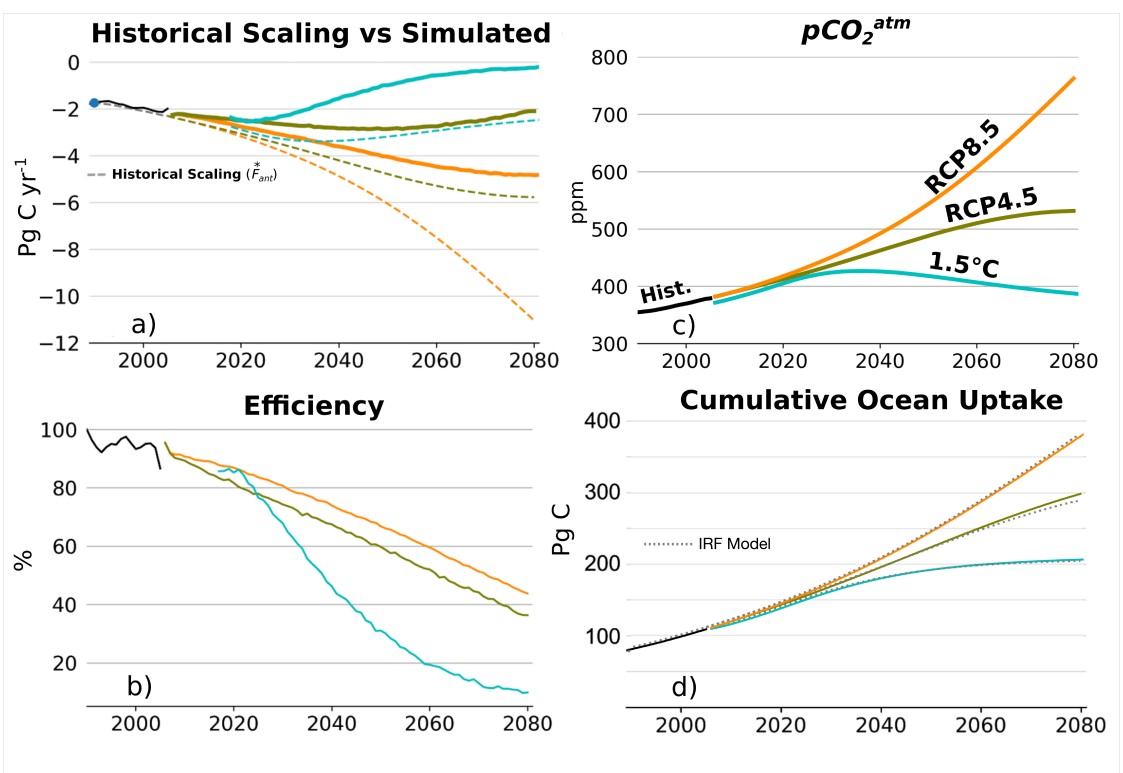

**Figure 2.** (a) Historical scaling of ocean anthropogenic carbon uptake ($\overset{*}{F}_{ant}$; dotted lines) and CESM anthropogenic carbon uptake ($F_{ant}$; solid lines) for three scenarios (1.5°C, RCP4.5, and RCP8.5). Negative indicates atmospheric anthropogenic carbon removal. (b) Efficiency of the global ocean sink for the three scenarios from CESM (Equation 5). (c) $pCO_2^{atm}$ for both CESM and IRF model. (d) Total anthropogenic carbon accumulation in CESM (solid lines) and in the IRF model (dotted lines). Flux and efficiency from 2006-2017 are not shown for 1.5°C scenario due to ocean adjustment to $pCO_2^{atm}$ forcing (see Methods 2.2; Figure S2).

Relative to the scenarios with emission mitigation (1.5°C and RCP4.5), the RCP8.5 scenario features a consistent spatial pattern of the air-sea flux of anthropogenic carbon (Figure 1, top row). The primary change over time is an amplification of magnitude, with the highest flux intensity occurring in 2080.

Global-mean anthropogenic carbon fluxes across the air-sea interface are greatest in RCP8.5, and lowest in 1.5°C (Figure 2a). In the RCP4.5 scenario, the air-sea flux of anthropogenic carbon peaks in 2050, and then gradually declines. In the 1.5°C scenario, ocean anthropogenic carbon uptake peaks in 2020, and is almost zero by 2080. In all scenarios, the ocean anthropogenic carbon inventory increases through 2080 (Figure 2d).

Extrapolation of the ocean anthropogenic carbon uptake based on the historical scaling ($\overset{*}{F}_{ant}$) is dependent solely on $pCO_2^{atm}$ (Equation 4). Lower $pCO_2^{atm}$ results in a lower estimate of ocean anthropogenic carbon uptake, and higher $pCO_2^{atm}$ results in a greater uptake estimate using the historical scaling. For all scenarios, CESM-simulated anthropogenic carbon uptake is far less than $\overset{*}{F}_{ant}$ (Figure 2a). Reduced uptake relative to $\overset{*}{F}_{ant}$ indicates that in the future, ocean anthropogenic carbon up-

take will be less efficient than for the "historical scaling" (Figure 2b). Efficiency remains greater than 90% from 1990 through 2010, but then declines under all future scenarios, with greater efficiency declines as emission mitigation increases. The efficiency decrease is approximately linear in RCP8.5 and RCP4.5, but exponential in the 1.5°C scenario. The 1.5°C scenario is the only scenario with negative $pCO_2^{atm}$ growth rates (Figure 2c).

## 3.2 Projected Changes in the Ocean Interior

Here, we analyze the evolution of the $C_{ant}$ vertical gradient by applying the historical scaling (Equation 6) to CESM's global-mean vertical profile of anthropogenic carbon ($C_{ant}(z)$). In Figure 3 and 4, deviations from the historical scaling are quantified as $C_{ant}(z) - \overset{*}{C}_{ant}(z)$. Weakening of the vertical $C_{ant}$ gradient reduces the strength of physical removal of anthropogenic carbon to depth and reduces the accumulation of $C_{ant}$ in the surface ocean (Equation 12). Wherever $C_{ant}(z) > \overset{*}{C}_{ant}(z)$, more carbon is stored at that depth than predicted by the historical scaling and the deviation is positive. If deviations are reduced at the surface relative to the interior, the vertical gradient is weakened, and thus ocean anthropogenic carbon uptake is less efficient.

With more rapid emission mitigation, globally average profiles reveal increasingly positive deviations from the historical scaling at depth (Figure 3). For RCP8.5 and RCP4.5, $C_{ant}(z)$ increases from 2020-2080 at all depths, but at the surface, $C_{ant}(z)$ increases less than $\overset{*}{C}_{ant}(z)$ (Figure 3a). In the RCP4.5 scenario, the anthropogenic carbon below 200m is greater than $\overset{*}{C}_{ant}(z)$ (Figure 3b), while in the RCP8.5 scenario it is lesser (Figure 3a). In both RCP8.5 and RCP4.5, the increase in anthropogenic carbon is surface-intensified. The resulting enhanced vertical gradient allows for increased downward physical transport of $C_{ant}$, and thus increased ocean anthropogenic carbon uptake (Equation 12). However, the enhancement of the vertical gradient is not as strong as the historical scaling would suggest.

In the 1.5°C scenario, the largest change from 2020 to 2080 in $C_{ant}(z)$ is at depth; at the surface, anthropogenic carbon decreases less significantly (Figure 3c). This leads to a much weaker vertical gradient, weaker vertical transport, and thus a reduced ocean anthropogenic carbon uptake. The surface loss of anthropogenic carbon is a short-term response to declines in $pCO_2^{atm}$ that begin in 2036, while the increase in $C_{ant}$ at depth is attributable to the long-term increase in $pCO_2^{atm}$ relative to preindustrial times, and the movement of this signal into the upper ocean through processes such as mode water formation (Bopp et al., 2015; Iudicone et al., 2016; Toyama et al., 2017).

The signals found in $C_{ant}(z)$ can also be identified in zonal-mean sections from CESM (Figure 4). In the RCP8.5 scenario (Figure 4, top row), the surface layer exhibits the strongest negative deviation from the historical scaling, but there is no positive deviation in the interior. The negative deviation is seen in deep waters between 25°N and 60°N, and also in the bowls of the northern and southern subtropical gyres. The negative deviation grows from 2020-2080, and appears to propagate into the ocean interior with NADW.

In the RCP4.5 scenario, the surface layer exhibits a growing negative deviation (Figure 4, middle). The negative surface deviation spans from the southern to the northern end of the zonal mean section. In the interior, however, there is a growing positive deviation. The positive deviation occurs because the ocean interior is not in contact with the atmosphere and thus the

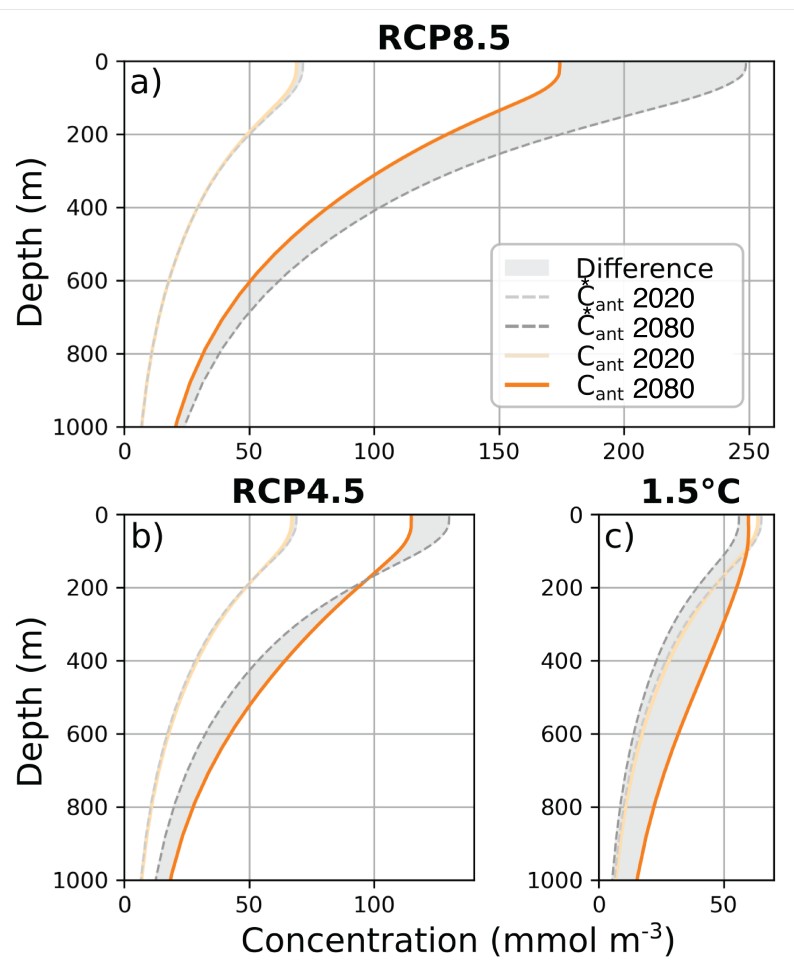

**Figure 3.** CESM global-mean anthropogenic carbon profiles ($C_{ant}(z)$) (orange, solid), and profiles of $\overset{*}{C}_{ant}(z)$ (gray, dashed), for the (a) RCP8.5 scenario, (b) RCP4.5 scenario, and (c) 1.5°C scenario. The shaded region between the dashed and solid lines indicates the deviation from the historical scaling. Light lines are for 2020 and dark lines are for 2080. The shaded region between the lines is shown for zonal mean sections in Figure 4.

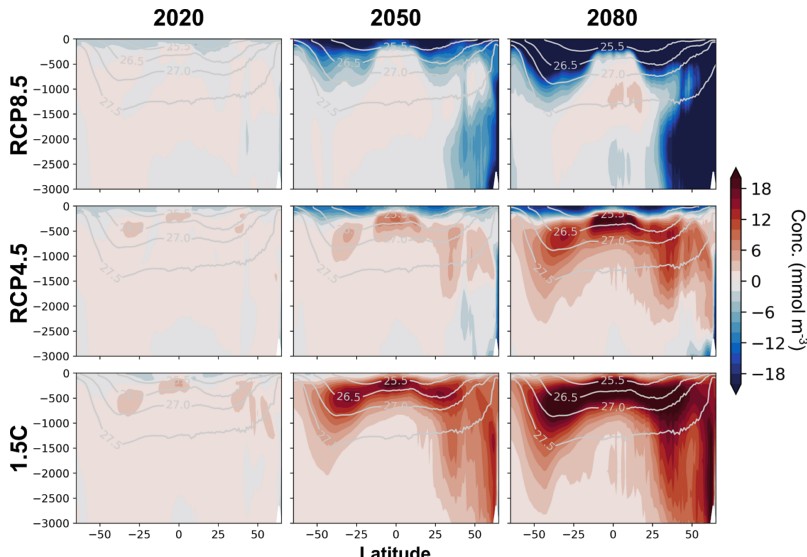

**Figure 4.** Ocean component model output of the global zonal mean deviation of anthropogenic carbon concentration (mmol m$^{-3}$) from the historical scaling of anthropogenic carbon ($C_{ant} - \overset{*}{C}_{ant}$). Rows and columns same as Figure 1. Positive regions indicate faster carbon accumulation than historical scaling, negative regions indicate slower accumulation. Contour lines are surfaces (kg m$^{-3}$).

ocean circulation is circulating $C_{ant}$ set by the $pCO_2^{atm}$ of prior decades. In other words, there is a lagged interior response to RCP4.5 in which $pCO_2^{atm}$ growth gradually slows (Figure 2c).

The 1.5°C scenario features even larger positive deviations from the historical scaling occurring throughout the thermocline (Figure 4, bottom row). As for RCP4.5, this occurs because the rapid slowdown of $pCO_2^{atm}$ is not immediately communicated to the interior. As thermocline waters outcrop in the equatorial Pacific and middle to high latitudes, they drive a source of anthropogenic carbon to the atmosphere (Figure 1).

### 3.3 Drivers of Simulated Changes in Efficiency

The IRF model reasonably replicates the cumulative ocean uptake of CESM (Figure 2d), supporting the assumption of constant circulation and the use of parameterized chemistry in the IRF. The IRF model can be manipulated for our sensitivity experiments (Table 1). With these experiments, a deeper mechanistic understanding of the changes in ocean carbon uptake efficiency simulated by CESM can be developed.

Over the historical period (1920-2005), accumulation of carbon ($\Delta C_{total}$) is nearly identical to the historical scaling (Figure 5). This is consistent with previous findings of the ocean sink being slightly less the theoretical prediction of the historical scaling (Raupach et al., 2014).

Under RCP8.5, the ocean absorbs 385 Pg anthropogenic carbon through 2080 (Figure 5, top, black line), approximately 2.5 times the present-day anthropogenic carbon inventory (160-166 Pg$C_{ant}$; DeVries, 2014). Due to the fact that ocean chemical capacity changes in the future, uptake is reduced significantly, -233 Pg$C_{ant}$ from 2020 to 2080 from what it would be if the

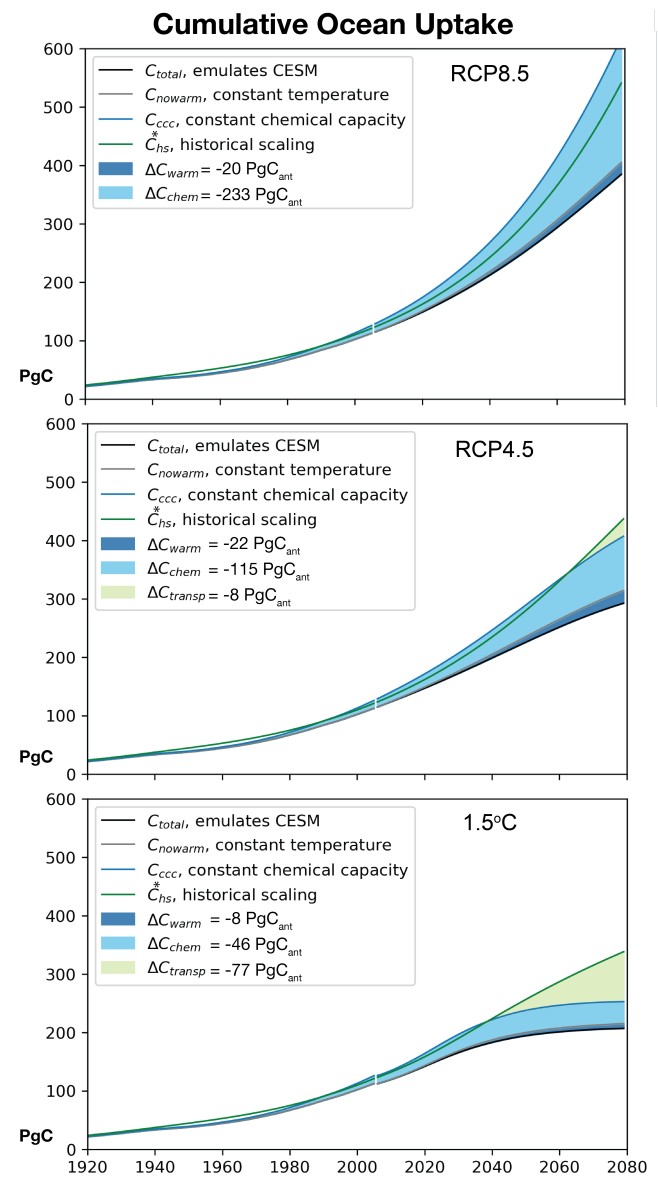

**Figure 5.** Cumulative ocean anthropogenic carbon uptake (Pg$C_{ant}$) in the IRF model; historical and for three future scenarios. The green line is the historical scaling ($\overset{*}{C}_{hs}$). The dark blue line is the IRF model simulation of constant chemical capacity with no impact of warming on solubility and inorganic carbonate chemistry ($C_{ccc}$). The gray line is the IRF simulation with no impact of warming on solubility and inorganic carbonate chemistry ($C_{warm}$). The black line is the IRF model simulation that includes all effects ($C_{total}$, variable chemical capacity and warming impacts on solubility and inorganic carbonate chemistry); this model replicates closely the cumulative carbon uptake of CESM (Figure 2d). Light green shading represents the residual, estimating the decrease in uptake related to vertical $C_{ant}$ transport ($\Delta C_{transp}$) for RCP4.5 and the 1.5°C scenario. Light blue shading represents decreases in uptake related to chemical capacity ($\Delta C_{chem}$). Dark blue shading indicates the decrease due to warming impacts on solubility and inorganic carbonate chemistry ($\Delta C_{warm}$). For each scenario, the carbon uptake from 2020 to 2080 is indicated in the label, with negative indicating loss relative to the total potential uptake.

buffer factor were to remain constant (light blue shade). In addition to this limit on uptake due to chemistry, there is a small additional reduction due to warming, -20 Pg$C_{ant}$ (dark blue shade). $C_{ccc}$ is substantially greater than the historical scaling (green line), cumulatively by 2080 causing a 98Pg$C_{ant}$ greater sink. This indicates that if the ocean were to have a fixed chemical capacity and experience no warming, it would be a substantially larger sink than estimated by the historical scaling. Exceeding the historical scaling is consistent with the RCP8.5 $pCO_2^{atm}$ having a trajectory that exceeds an exponential after 2012 (Figure S3). As in previous studies of climate-carbon feedbacks (Randerson et al., 2015; Schwinger and Tjiputra, 2018), we find that buffering is primarily responsible for limiting the ocean carbon uptake under high emissions scenarios through 2080, and that warming plays a secondary role.

Under RCP4.5, the ocean absorbs 292 Pg anthropogenic carbon (Figure 5, middle, black line) through 2080. Cumulative uptake predicted by the historical scaling is slightly lower than the constant chemical capacity experiment ($C_{ccc}$) through 2060. The transport effect (light green shade), $\Delta C_{transp}$, only appears after 2060 and has only a small impact, -8 Pg$C_{ant}$ cumulatively through 2080. This combines with the stronger $\Delta C_{chem}$ effect (-115 Pg$C_{ant}$), and the impact of warming on solubility and inorganic carbonate chemistry ($\Delta C_{warm}$ = -22 Pg$C_{ant}$). In total, the ocean carbon sink is reduced by a total of 33% from the historical scaling, due mostly to carbonate chemistry.

Under the 1.5°C scenario, the ocean absorbs 207 Pg anthropogenic carbon (Figure 5, bottom, black line) by 2080. $\Delta C_{chem}$ reduces uptake from the historical scaling (-46 Pg$C_{ant}$ in 2080), and the additional impact of warming is -8 Pg$C_{ant}$. The weaker $\Delta C_{chem}$ effect than in the other scenarios is consistent with the ocean taking up far less anthropogenic carbon in this scenario (Figure 2d). In contrast to the other scenarios, $\Delta C_{transp}$ (light green shade) is the dominant factor that reduces carbon uptake from the historical scaling, accounting for -77 Pg$C_{ant}$. The strongly reduced vertical gradient of anthropogenic carbon (Figure 3, 4) results in reduced vertical transport from surface to depth (Equation 12). For the 1.5°C scenario, the ocean carbon sink is reduced by 39% from the historical scaling, with over half of this change due to vertical $C_{ant}$ transport and the remainder due mostly to inorganic carbonate chemistry.

## 4 Discussion

### 4.1 Drivers of Future Efficiency Declines

We use the CESM and an IRF model that emulates the CESM's global-mean behavior to assess the mechanisms of future change in the ocean carbon sink as dependent on the future $pCO_2^{atm}$ and ocean internal accumulation of anthropogenic carbon ($C_{ant}$). We show that the efficiency of ocean carbon uptake, i.e. how closely ocean carbon uptake follows the observed proportionality between uptake and atmospheric $CO_2$ (the "historical scaling"), will be substantially reduced in the future under all projected emissions scenarios. However, the controlling mechanisms for these change will depend on the scenario. Our findings are consistent with theory (Raupach et al., 2014) and past idealized modeling studies (Zickfeld et al., 2016; Schwinger and Tjiputra, 2018).

The dominant mechanisms of efficiency decline differ across the three scenarios for future $pCO_2^{atm}$. With above-exponential growth of $pCO_2^{atm}$ in RCP8.5, the strong increase of $C_{ant}^{ML}$ concentrations causes a reduced chemical capacity that dominates

the reduction in efficiency (Figure 5, top). At the same time, a strong surface to depth gradient of $C_{ant}$ is maintained (Figure 3a, 4), supporting continued downward transport of carbon to the ocean interior (Equation 12). In RCP4.5, chemical capacity is also the dominant driver of the reduced sink, but a weakened vertical $C_{ant}$ gradient allows the transport effect to begin to play a role after 2060 (Figure 5, middle). In the 1.5°C scenario, a significant weakening of the vertical gradient of $C_{ant}$ (Figure 3c) dominates the reduction in efficiency (Figure 5, bottom).

With emission mitigation, the vertical gradient of $C_{ant}$ does not immediately adjust to the trajectory of $pCO_2^{atm}$. Anthropogenic carbon accumulation from 2020-2080 is greatest in the thermocline; a behavior that has been identified in other simulations of strong mitigation (Tokarska et al., 2019). This accumulation weakens the vertical gradient of $C_{ant}$ (Figure 3, 4) and reduces the downward transport of $C_{ant}$. The bolus of anthropogenic carbon held at depth creates a "back-pressure" that resists additional flow of anthropogenic carbon into the interior. As emissions are mitigated, the back-pressure grows (Figure 3-5). As the magnitude of the air-sea flux of anthropogenic carbon is fundamentally limited by the rate of surface to depth transport of $C_{ant}$ (Graven et al., 2012), slower removal to depth results in a reduced carbon uptake from the atmosphere.

Regionally, ocean circulation impacts $pCO_2^{ocn}$ through advection and watermass transformation (Bopp et al., 2015; Toyama et al., 2017). Advection returns to the surface waters that have already absorbed $C_{ant}$, and if the $pCO_2^{atm}$ is falling when these waters remerge, the surface ocean carbon content will exceed the atmosphere and outgassing will occur. For the 1.5°C scenario, this occurs in the equatorial Pacific, subpolar and mid-latitude North Atlantic, SAMW outcrop region, and the Kuroshio (Figure 1, bottom). However, waters of the subtropics are renewed with waters that are shallower than where significant $C_{ant}$ accumulation occurs, and surface waters of the Southern Ocean are renewed with deep waters without any $C_{ant}$. Thus, in some parts of the subtropics and Southern Ocean, $C_{ant}$ uptake continues even with emissions mitigation while there is $C_{ant}$ outgassing elsewhere. Particularly under aggressive emission mitigation, substantial shifts in the regional patterns of air-sea carbon fluxes can be expected. These shifting patterns will need to be taken into account when planning for carbon cycle monitoring and diagnosis (Peters et al., 2017).

Whether before or after 2080, eventually emissions will decline either due to purposeful mitigation efforts or to the exhaustion of fossil fuel reservoirs. For example, under the RCP8.5 scenario, emissions would be flat from 2100 to 2150 and then decline dramatically (van Vuuren et al., 2011). The back-pressure effect due to the vertical gradient of $C_{ant}$ in the ocean will be delayed as long as $pCO_2^{atm}$ is rapidly growing, but it will eventually play a role in reducing the ocean carbon sink. The longer mitigation is delayed, the greater the load of $C_{ant}$ in the thermocline will be, and thus the back-pressure effect will be larger in magnitude and temporal duration. More climate simulations extending beyond 2100 are needed to quantify the back-pressure effect under all scenarios. Limiting emissions now makes it possible to reduce the eventual magnitude of the back-pressure effect and also to avoid the ocean chemistry changes that will additionally slow future ocean carbon uptake (Figure 4, 5).

## 4.2 Validity of the Model Representations of Ocean Physics

The back-pressure from anthropogenic carbon at depth is an unavoidable consequence of emission mitigation. How long the ocean will remain a net sink depends on the strength of the back-pressure effect, which depends on the how fast anthropogenic carbon is removed from the surface ocean to depth. This makes the fidelity of the ocean physics represented in the CESM, and

then fit with the IRF model, very important. The IRF model represents multiple physical processes that remove carbon to depth as the decay of of a surface flux over time. This decay has been set (Section 2.3) so as to mimic advective, eddy-diffusive and watermass transformation processes occurring in CESM. Iudicone et al. (2016) show that advection and diabatic processes in watermass transformation are most important to the storage of $C_{ant}$ in the mode waters of the upper ocean.

For the historical period, global-mean air-sea fluxes and anthropogenic carbon storage are not substantially different across three-dimensional models, despite these models having substantial differences in the ocean circulation (Winton et al., 2013; McKinley et al., 2016; Bronselaer and Zanna, 2020; Hauck et al., 2020). This result is consistent with the external forcing from the growth of atmospheric pCO$_2$ being the overwhelming driver of the historical sink (McKinley et al., 2020). Looking forward to a changing trajectory of the atmospheric boundary condition, uncertainties in the ocean circulation, as indicated by the spread of model predictions for ocean heat uptake (Bronselaer and Zanna, 2020), may become important. For this study, we focus on evaluating the mechanisms in operation in CESM, but if we were emulating the ocean component of a different ESM, findings will likely be quantitatively different. Though assuming that change in the ocean circulation has a small impact on the carbon cycle prior to 2080 is consistent with the behavior of the CESM under RCP8.5 (Randerson et al., 2015), this may not be hold true for other ESMs or the real Earth. A valuable direction for future work will be to evaluate the spread in predictions for both inorganic carbonate chemistry and vertical transport effects.

## 5 Conclusion

Atmospheric CO$_2$ has grown exponentially over the industrial era, and so has ocean anthropogenic carbon concentration at depth (DeVries, 2014; Gruber et al., 2019). Under an exponential forcing regime, ocean anthropogenic carbon uptake also grows exponentially. Since these conditions have held over the historical era, the ocean sink has historically maintained a high efficiency. In future scenarios, regardless of the degree to which emissions are mitigated by 2080, efficiency of ocean anthropogenic carbon uptake will decline. We show that the mechanisms of this decline will differ depending on the degree of mitigation. In the RCP8.5 and RCP4.5 scenarios, reduced buffer capacity explains most of the loss in ocean sink efficiency through 2080. With strong mitigation in the 1.5°C scenario, the loss of efficiency is due more to the effect of vertical transport of $C_{ant}$, which explains more than half of the efficiency loss.

Change in the vertical anthropogenic carbon concentration gradient is responsible for the changing impact of vertical transport of $C_{ant}$ on the ocean sink. When emissions are mitigated and the growth in $pCO_2^{atm}$ slows, the surface ocean carbon content responds rapidly. However, the ocean interior anthropogenic carbon concentration response lags the surface response. Below 100m in the 1.5°C scenario, anthropogenic carbon concentration increases from 2020-2080; but above 100m, the anthropogenic carbon concentration begins to decrease starting in 2038, just two years after the maximum $pCO_2^{atm}$ of 437ppm is achieved. The downward anthropogenic carbon concentration gradient is greatly reduced and there is less effective downward transport of $C_{ant}$. Ocean anthropogenic carbon uptake is limited by the removal of anthropogenic carbon from surface to depth (Graven et al., 2012). As the vertical gradient changes in the future, this transport is reduced and there will be less future uptake relative to what occurred at the same $pCO_2^{atm}$ concentration in the historical period (Schwinger and Tjiputra, 2018).

The upper ocean circulation will play a critical role in the efficiency of the ocean carbon sink as the $pCO_2^{atm}$ growth rate

begins to slow. Current ocean model estimates of the ocean carbon sink agree well for the global-mean carbon uptake (Hauck et al., 2020) and future estimates under high emissions do not diverge substantially through 2100 (Arora et al., 2013). However, these simulations do diverge in their predictions of recent and near-future heat uptake, a process that is much more dependent on circulation details (Bronselaer and Zanna, 2020). This suggests that for scenarios of aggressive emission mitigation, model predictions of the ocean carbon sink may diverge much more than in the high emissions scenarios that have been the primary

focus to date (Friedlingstein et al., 2013; Randerson et al., 2015). Next steps will be to determine how much these simulations do diverge, and then to work to reduce these uncertainties. The ocean carbon sink plays a critical role in the global carbon cycle and the climate. Accurate predictions of its magnitude under all plausible future scenarios for $pCO_2^{atm}$ are essential.

*Code and data availability.* The code used to run the IRF model is provided by the authors in a GitHub repository (https://qoccm.readthedocs.io/en/latest/). Raw output from the coupled ocean model simulations can download from NCAR's Earth System Grid (https://www.earthsystemgrid.

org/).

## Appendix A: Ocean Carbon Cycle Model Carbon Chemistry for the Impulse Response Function Model

The $pCO_2^{ocn}$ of the IRF ocean carbon cycle model is calculated using the empirical fit to the a solution of the carbonate system equations by Joos et al. (2001). We use a fitted solution for two reasons. First, when variables other than temperature and carbon are held constant, using the full carbonate system equations provides no additional accuracy. Second, the concentration

scenarios used in CMIP5 (RCP4.5, RCP8.5) with which we wish to be consistent were generated using the same IRF model with the same representation of ocean chemistry.

$$pCO_2^{ocn} = [pCO_2^{ocn,PI} + \delta pCO_2^{ocn}(C_{ant}, T_{pi})]exp(\alpha_T \delta T) \qquad (A1)$$

Where $pCO_2^{ocn,PI}$ is the preindustrial global-mean $pCO_2^{ocn}$. The response of $pCO_2^{ocn}$ to warming is parameterized as an exponential function as in Takahashi et al. (1993), with $\alpha_T$ set to 0.0423 $K^{-1}$. The carbonate chemistry that determines

$\delta pCO_2^{ocn}$ given anthropogenic carbon ($C_{ant}^{ML}$) is parameterized assuming a fixed ocean alkalinity of 2300 $\mu$mol kg$^{-1}$ and the preindustrial temperature, $T_{pi}$, based on an empirical fit to carbonate system calculations (Equation A24; Joos et al. (2001))

$$\delta pCO_2^{ocn}(C_{ant}, T_{pi}) = C_{ant}[A1 + C_{ant}(A2 + C_{ant}(A3 + C_{ant}(A4 + C_{ant}A5)))] \qquad (A2)$$

With coefficients :

$$A1 = (1.5568 - 1.3993 \times 10^{-2} \times T_{pi}) \tag{A3}$$

$$A2 = (7.4706 - 0.20207 \times T_{pi}) \times 10^{-3} \tag{A4}$$

$$A3 = -(1.2748 - 0.12015 \times T_{pi}) \times 10^{-5} \tag{A5}$$

$$A4 = (2.4491 - 0.12639 \times T_{pi}) \times 10^{-7} \tag{A6}$$

$$A5 = -(1.5468 - 0.15326 \times T_{pi}) \times 10^{-10} \tag{A7}$$

*Author contributions.* Both authors contributed to the discussion and the writing of the paper

*Competing interests.* The authors declare that no competing interests are present

*Acknowledgements.* We would like to thank the National Science Foundation Division of Ocean Sciences (OCE-1948624 and OCE-1818501), NASA Earth Sciences Division (NNX/17AK19G), and Columbia University for funding this research. We would also like to thank Jared Lewis and Bodeker Scientific for sharing their pySCM (python Simple Climate Model, https://github.com/bodekerscientific/pyscm/) code on GitHub. This code provided the basis the IRF ocean carbon cycle model presented here. We acknowledge the CESM Large Ensemble Community Project and supercomputing resources provided by NSF/CISL/Yellowstone. This material is based upon work supported by the National Center for Atmospheric Research, which is a major facility sponsored by the National Science Foundation under Cooperative Agreement No. 1852977. We thank all the scientists, software engineers, and administrators who contributed to the development of CESM.

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
