# Peer review of "Ocean Carbon Uptake Under Aggressive Emission Mitigation"

_Biogeosciences, 2020_

## Referee Comment (RC1) · Anonymous Referee #1 · 6 Aug 2020

**General Comments**

The manuscript by Ridge and Galen analyses the factors behind the decreasing efficiency of oceanic anthropogenic carbon ($C_{ant}$) uptake in three different ensembles of climate model model runs with the community earth system model, until the end of the current century. The ensembles differ in their forcing of the carbon cycle and range from a scenario with continuing strong emissions, RCP8.5, an intermediate emissions scenario (RCP4.5) and a scenario where the radiative forcing from the emissions remains so low that the 1.5°C target set in the Paris agreements is reached. Of these, the last and most desirable scenario shows the strongest decrease in the efficiency of the ocean to take up $C_{ant}$. A priori this is to be expected because a decrease in carbon

emissions will lead to a less strong pressure on the ocean to take up more carbon.

It is known that, globally, the efficiency of $C_{ant}$ uptake in the ocean will decrease through several mechanisms. One effect is a purely chemical one, caused by the declining buffering capacity of seawater with increasing $pCO_2$, together with the temperature effect on carbonate chemistry. A second is what the authors call the "carbon gradient effect", and has to do with the rate of incrase of atmospheric $pCO_2^a$: The surface ocean $pCO_2^o$ tends to equilibrate with the atmosphere, but at the same time anthropogenic carbon is moved into the interiour of the ocean by diffusion and advection, working against this trend. Any slowing of the $pCO_2^a$ increase therefore will lead to lessening of the carbon gradient between atmosphere and ocean, reducing the uptake (Raupach et al., 2014).

The manuscripts investigates the relative role of these two processes by fitting a one-dimensional reaction-diffusion model for ocean carbon uptake to the evolution of ocean $C_{ant}$ in the ensemble averages over the historical period. Using the one-dimensional model then one can easily separate the two above-mentioned effects in the scenario runs.

This approach is an elegant way to help understanding how future uptake of carbon in the ocean will evolve and merits publication in Biogeosciences. But I would suggest two larger changes, before the manuscript should be accepted:

The first is that the manuscript could be significantly shortened. One candidate for shortening are the many repetitions of the main results in chapter 3, 4 and ultimately 5. Especially the discussion subchapter 4.1 reads like a lengthy repetition of the main results, rather than a true discussion. The other candidate for shortening is that the manuscript is in several places much too verbose to explain fairly standard calculations and mathematical techniques. I list those in more detail under specific comments.

The second is that the manuscript hardly relates the results obtained with some of the previous literature: especially the earlier studies on carbon and climate feedbacks on

ocean carbon uptake are cited in the introduction, but their results are later not related to again. Feedback analysis of the last CMIP5 model runs have e.g. shown a climate sensitivity of ocean carbon uptake of -8 $\pm$ 3 GtC K$^{-1}$ (e.g. Friedlingstein, 2015). What part of this feedback would be included in which effect in the analysis performed here? While I see that the methodologies in the feedback analysis and the methodology used here may not be easy to concile, it would at least be useful to discuss how they are related, and whether the results are compatible. The authors also state somewhat in passing (lines 348-349) that changes to the ocean circulation do not seem to play a large role in C$_{ant}$ uptake. This should also be discussed somewhat more.

**Specific Comments**

The nomencalture of anthropogenic carbon is handled quite confusingly, with C$_{ant}$ meaning very different things in different places in the manuscript without proper distinction. In some places, C$_{ant}(z,t)$ is used to descibe the time-and depth-dependent anthropogenic carbon in the 1-d model, in other places, either the depth or time-dependency is left away (e.g. line 142). In equation 7, then again C$_{ant}$ means the surface ocean concentration only. I suppose that is also, what is meant on the left-hand side of equation 8, while on the right hand side the depth- and time-dependent field is meant. In that form, the equation cannot hold in the interior of the ocean. I suggest to clarify what is meant in each instance e.g. by adding additional superscripts, e.g. C$_{ant}^{ML}$ for the mixed-layer C$_{ant}$.

I think readers are able to understand the linear equation 4 and its consequence equation 5 without the lengthy explanation in lines 114 to 124.

A similar statement holds for the explanation of the impulse response function 7 from lines 179 to 195; this is a fairly standard mathematical technique and does not need to be explained in so much detail.

[Figure]

An important point in the methods chapter is line 220 to 222, where it is stated that the relation between $C_{ant}^{ML}$ and $\delta p CO_2^{ocn}$ contains the effects of changing buffer factor and of changing temperature. Here would be a good place to discuss how the decomposition of effects made here is related to the more traditional feedback analysis that results on the two feedback factors $\beta$ and $\gamma$. I must admit that I wondered why it is necessary to have fitted relation between the two (Appendix B), rather than using standard carbonate chemistry and calculating one from the other, assuming constant alkalinity.

Line 314: "Assuming ocean circulation remains constant" To what extent is that assumption justified?

I do not completely understand, how the fields shown in Figure 4 have been calculated. Is this the zonal average of the 3-d model output minus the expected profile from the 1-dimensional model? How does advection enter this picture?

Section 4.1 contains very little discussion and is in large parts a repetition of results. Then, in the end (lines 468-471), some other results on future $C_{ant}$ uptake are cited, but without giving any connection to what this study has shown. Is there a relation between these results, or is it just inserted here without too much meaning?

Line 410 ff: "We find ...": Is this really a finding, or not rather an assumption that went into the methodology?

Appendix A again explains the impulse response function, even repeats equation 7, and gives details about using it that can be found in many textbooks. I suggest to remove it (including Figures A1 and A2), or shorten it to not more than one paragraph.

Appendix B: The coefficients shown in equation B3 to B7 all have different units, but these are not shown.

**Technical Corrections**

Lines 297 and 298: should one of the mentioned scenarios be RCP8.5 rather than RCP4.5?

References: Many references give two web addresses for the cited papers, one being the http-form of the doi (I would rather have the doi without the `https://doi.org` in front), and the journal address. Are really both necessary?

**References**

Friedlingstein P. (2015): Carbon cycle feedbacks and future climate change. Phil. Trans. R. Soc. A 373: 20140421. doi: 10.1098/rsta.2014.0421

---

## Referee Comment (RC2) · Anonymous Referee #2 · 19 Aug 2020

This manuscript analyzes the efficiency of ocean carbon uptake, defined relative to an exponential scaling, for scenarios of different future CO2 concentrations (ranging from a worst case to a strong mitigation case). The authors use an Earth system model as well as a simple 1d-model of the ocean carbon cycle fitted to the ESM. The simple model is used to decompose uptake efficiency changes seen in the ESM simulations into different drivers. The results demonstrate the strong decline in efficiency (according to the chosen metric) in the strong mitigation scenario, which is driven by changes in the vertical gradient of anthropogenic carbon in the ocean.

Although the basic principle of declining uptake efficiency under strong mitigation is quite intuitive, the analysis and decomposition into drivers provided by this manuscript is definitely of interest to the readers of Biogeosciences. However, the manuscript is

(mainly in the introduction and the methods section) not well structured, suffers from imprecise language, and is generally not very concise. Assumptions are often not clearly stated. As the manuscript stands, it does not meet the high standards of Biogeosciences. I give a few examples below (point 1), but my concerns are not limited to these examples. I urge the authors to go carefully through their text and rewrite (with a focus on conciseness) the introduction and the methods section. I have also a few other, more scientific concerns, which I also list below.

Major points:

1) Structure and conciseness of the manuscript (mainly introduction and methods):

*The concept of an exponential growth of emissions leading to a constant sink rate (under assumptions) is quite central in this work, and it needs to be introduced and put into context. Currently this concept is first mentioned in passing in line 44. It needs to be introduced to the reader before the sink rate is mentioned.

*line 50-53: " Nearly every nation..." I don't see that this sentence adds anything new here, consider deleting.

*The authors claim that "In the RCP8.5 scenario (Meinshausen et al., 2011), pCO2_atm increases exponentially..." (line 71). To me it is unclear to which degree the RCP8.5 emissions or concentraions can be approximated by an exponential, but this is not very relevant to this study either (since the baseline is an idealized exponential growth). RCP8.5 is the outcome of an advanced modelling exercise, so the emissions are not strictly exponential.

*Throughout the manuscript, the authors mention exponential historical emissions. It should be made clear that this is an idealization (e.g. by saying "roughly exponential" or similar). This is already the case in some places but missing in others.

*lines 54-67: Again, all this could be much more concise. The feedback studies mentioned are considering a single (exponential) concentration pathway, so they cannot

quantify an uptake efficiency for different emission pathways. (And yes, in addition to this, they cannot quantify the contribution of a changing buffer factor, either).

*line 74-78: Consider moving this up to lines 40-50 where k_s in general is introduced.

*lines 102-106: This text is not necessary, we do not need a summary of subsections at the beginning of a section. Please consider removing. The same applies for lines 271-277.

*line 108: What is the first sentence of 2.1 supposed to tell us? Just start with "The efficiency metric (eta) used here is defined as k_m ..."

*The text explaining equation 4 (lines 112-118) can be shortened substantially: "The historical scaling for ocean C_ant uptake (F_ant) is defined as: (equation 4). The overset "*" indicates that a variable has been extrapolated using the historical scaling. Here, we diagnose F_ant(1990) from the CESM large ensemble simulations. For example...".

*Delete unnecessary words, e.g. "mathematically".

*Lines 129-130: "While k_M remains constant,...". This does not reflect the logic of this manuscript. The authors use the exponential scaling to define a baseline against which simulated quantities are compared. The actual k_M is not and does not need to be constant for this exercise.

*Line 144: "The CESM provides a realistic simulation of the response of the ocean carbon cycle to climate change." What do the authors mean by "realistic"? I would suggest to delete this sentence.

*Section 2.3: Impulse response functions are a well established tool in climate modelling. It is useful to give a short explanation for those readers that are not familiar and highlight those aspects that relevant for this study, but otherwise the authors should refer the reader to the literature and shorten section 2.3 substantially. Likewise the Appendix A is not necessary. The most important assumption of IRFs is constant circulation. The most important aspect of the Joos-IRF (hidden in the Appendix A) is

the fact that, contrary to atmospheric IRFs, the mixed layer IRF take ocean carbon chemistry (including changes in SST) into account.

*Throughout section 2, it is unclear what the symbol $C_{ant}(t)$ is supposed to denote. In equation 7 it is the anthropogenic carbon content of the mixed layer, but otherwise $C_{ant}$ (often) seems to denote the total ocean anthropogenic carbon. The authors also frequently use the expression "$C_{ant}$ air-sea flux", which I would suggest to replace by "air-sea flux of anthropogenic carbon" (and use $F_{ant}$ as a short form of this if necessary).

*Section 2.3: The remaining description of the 1d-model is verbose and confusing. Apparently, the authors "extend" the mixed layer IRF downward by "plugging" a diffusion equation under the mixed layer IRF? Yes, the downward flux can be determined "by residual", but then how are the profiles of $C_{ant}(z)$ calculated?

*Section 2.5: Equation 12 can be derived by assuming $F_{ant}$ = $F_{ant}(pCO2atm(t),pCO2ocn(t))$. Then, later, it is additionally assumed $pCO2ocn=pCO2ocn(C_{ant},T)$. This should be made clearer. (Again $C_{ant}$ here means surface $C_{ant}$).

*Section 2.5: The equation del $F_{ant}$/del pCO2atm = del $F_{ant}$/del pCO2ocn, is this based on Equation 10? Then a minus sigh is missing. Also, the dependency of the transfer velocity on temperature is neglected in this step. How does it follow from Equation 12 that "The pCO2ocn closely follows pCO2atm, and the sign of their growth rates is the same"?

*Section 3.1: If a paragraph begins with "In the RCP4.5 scenario, changes to the spatial pattern lie somewhere between RCP8.5 and the 1.5C scenario" both scenarios should have been discussed already. This is not the case here.

As pointed out above, this list is not exhaustive.

2) In my opinion, the term "historical scaling" used by the authors is misleading or at

best confusing. The sink rate has not been constant over a substantial part of the historical period in observations (Raupach et al. 2014, cited) as well as in the model experiments used in this study (Fig S1). If the authors wish to replace the previously used term "transient steady state", why not just saying what it is, e.g. "exponential scaling" (or something similar that does not refer to the historical period)?

4) Choice of time periods: The time period 1920-2006 is not the "historical period". From a CMIP5 perspective this would be 1850-2006. Why do the authors choose 1920 as a starting year? Likewise, why do the authors not use the last 20 years of the scenarios, which would be most interesting period in the mitigation scenarios?

5) 1d-model evaluation: The space saved by shortening Section 2 could be invested in presenting a brief evaluation of the (full) 1d-model compared to CESM. How well does the fitted 1d-model reproduce $F_{ant}$ in the 3 different scenarios? More important, how well are vertical profiles of $C_{ant}$ simulated? To me it is not given that the 1d diffusion model has skill in reproducing the CESM global mean $C_{ant}$ profiles for all scenarios, but this is the basis of the analysis of the "gradient effect".

6) Section 3.4: This section is not easy to follow. What is the main point here? I guess it is the fact that the ocean uptake in the strong mitigation scenario after 2040 is maintained by the ocean through continuous downward mixing (otherwise the surface ocean would start outgassing because pCO2atm declines already). Could the authors please add some easy to understand explanations here? Further, related to my point 5) above, how realistic is this process simulated by a 1d-model? In reality we would have upwelling of waters that have been last in contact with the atmosphere in pre-industrial times, that can potentially sustain ocean uptake even under declining CO2, but this is not the case in the 1d-diffusive model. Here the processes must be different. Could the authors please comment on this?

7) The authors should appropriately acknowledge the tremendous work of hundreds of scientists and engineers that made the CESM simulations possible. The

NCAR CESM website specifically asks authors to do this, this is not optional (http://www.cesm.ucar.edu/publications/).

Minor points:

line 20-21: "The ocean has absorbed 39% of the CO2 from industrial era fossil fuel combustion and cement production (Friedlingstein et al., 2019). The rest of the CO2 remains in the atmosphere where it acts as the primary driver of climate change." Even if the authors seem to neglect emissions from land-use change the number 39% is not correct (it is 36% calculated from Friedlingstein page 1812), but more importantly, land use change should be included here, since it is an anthropogenic emission to the atmosphere (and is treated separately in the GCB). According to Friedlingstein et al. 2019, the ocean has absorbed 25% of the CO2 emissions from industrial era fossil fuel combustion, cement production, and land use change (Friedlingstein et al. 2019, page 1812).

line 35: "...one-dimensional diffusion models have been shown to be consistent with observations and complex models (Gnanadesikan et al., 2015; Oeschger et al., 1975)." Please consider adding "on a global scale" or similar.

line 54: "The reductions to efficiency that are attributable to a slowing pCO2_atm growth rate will be at least partially compensated by a decrease in the strength of the climate-carbon feedbacks". Climate-carbon feedbacks will decrease F_ant,M, so why would this compensate a decrease of k_M? Please clarify.

line 86-87: "... C_ant concentration at all points in space also follows the historical scaling" and "The amplifying effect..." It is unclear to me what is meant by "follows the historical scaling" and "amplifying". Please clarify.

line 176-178: The MAGICC model uses the Joos et al. 1996 IRF for the Princeton GCM, not the HILDA model as the authors do. So it would be correct to say "A very similar IRF has been used to..."

[Figure]

line 293: "...the RCP8.5 scenario features a consistent spatial pattern of the C_ant air-sea flux..." What does "consistent" mean here? Please consider rewording.

line 306: "The efficiency decrease is linear in RCP8.5 and RCP4.5, but exponential in the 1.5C scenario." The decrease is neither exactly linear in the RCPs nor exactly exponential in the 1.5 scenario. Please add "approximately" or similar.

line 408: "...i.e. how closely ocean carbon uptake follows the observed proportionality between uptake and atmospheric CO2". The proportionality is not exactly observed. Please be more precise here.

line 409: Again, in RCP8.5 pCO2atm might increase roughly exponential.

line 429-430: Here it should be mentioned that changes in ocean circulation due to warming are not consistently treated in this study, since they are present in CESM, but not in the 1d-model.

line 445-446: "Therefore, climate simulations extending beyond 2100 are needed to quantify the back-pressure effect in high emission scenarios." These simulations exist, the CMIP6 SSP5-8.5 scenario has an extension to 2300.

Technical:

line 37: "is described" -> "can be described"

line 67: delete "additional"

line 69: "We will compare..." -> "We compare..."

line 115, 117: delete "mathematically"

line 228-229: "With these experiments,..." I would suggest to reword this sentence: With these experiments we can decompose the total anthropogenic carbon uptake into contributions from...

line 297-298: The 3rd sentence seems to repeat the 1st sentence? Please consider

removing.

line 314: "...and the deviation is positive" This can be deleted.

line 459: "but" should read "such that"(?)

In figure captions: Replace "Output from the ocean component of the CESM of the global mean C_ant profiles..." by "CESM global mean ocean C_ant profiles..." (and similar in other figure captions).

————————————————

---

## Short Comment (SC1) · 27 Aug 2020

The study of Ridge and McKinley is to be commended for their investigation of mechanisms that contribute to controlling the net ocean uptake of carbon under future climate change. However, there are a few points that I think should be clarified in order that the study be more firmly anchored in previously published research.

Readers of this work should be informed however that Gnanadesikan et al. (2015) is not the definitive work on attribution of the mechanisms controlling the flux of $CO_2$ between the surface ocean and the interior. The study of Iudicone et al. (2016, SR) in fact considered this question of the relative importance of diapycnal transports and diffusive processs, and demonstrated for the primacy of diapcycnal transports as part

of the overturning circulation in controlling the formation of the ocean interior carbon reservoir

With that in mind, I would recommend that the mechanism highlighted in the manuscript n Equation 14 (ocean dynamics term as being diffusive) be described as an assumption based on the authors' interpretation of the previous study of Gnanadesikan (2015), rather than as a result or a conclusion. But it should also be stated that this interpretation was not in fact tested or fully asserted by Gnanadesikan (2015), I believe that they saw it more as a curiosity that a diffusive model can be tuned to give a result that matches what one finds with a forward model under very specific scenarios.

Putting all of this into more broadly resonant language, I think that it is commonly understood that mode water formation should not be understood to be a fundamentally diffusive process, and given the importance of mode waters for the uptake of anthropogenic carbon (of order half the global uptake) a dominantly diffusive uptake model would run counter to oceanographic observations and oceanographic theory.

I think that this needs to be very clearly communicated to the reader in order to firmly anchor this study in the context of broader oceanographic process understanding. I think that this will then strengthen the scientific presentation of the study.

---

## Author Comment (AC2) · 12 Sep 2020

[bg, manuscript]copernicus

**Response to Reviewer #1**

Thank you for your careful review of our manuscript, your suggestions have helped us improve our manuscript. We have shortened our manuscript by removing all of the text you identified in your detailed comments and we have removed Appendix A. Also, the detailed summary in section 4.1 has been removed and replaced with a much more succinct version that better links with the cited literature. Finally, we have added sentences to section 4.2 contextualizing our work with that

of the climate-carbon feedbacks literature. Our detailed responses can be found below.

**The nomencalture of anthropogenic carbon is handled quite confusingly, with Cant meaning very different things in different places in the manuscript without proper distinction. In some places, Cant(z,t) is used to describe the time-and depth-dependent anthropogenic carbon in the 1-d model, in other places, either the depth or time dependency is left away (e.g. line 142). In equation 7, then again Cant means the surface ocean concentration only. I suppose that is also, what is meant on the lefthand side of equation 8, while on the right hand side the depth- and time-dependent field is meant. In that form, the equation cannot hold in the interior of the ocean. I suggest to clarify what is meant in each instance e.g. by adding additional superscripts, e.g. CML ant for the mixed-layer Cant.**

We understand that our current naming convention is potentially very confusing. We have followed your suggestions and updated the symbols to the following:

| Name | Symbol | Units |
|------|--------|-------|
| Time, depth, and space dependent anthropogenic carbon concentration | $C_{ant}(x,y,z,t)$ | mmol m$^{-3}$ |
| Mixed layer $C_{ant}$ concentration | $C_{ant}^{ML}(t)$ | mmol m$^{-3}$ |
| Atmospheric anthropogenic carbon inventory | $C_{ant}^{ATM}(t)$ | Pg C |

**I think readers are able to understand the linear equation 4 and its consequence equation 5 without the lengthy explanation in lines 114 to 124.**

We agree and have updated this section

**A similar statement holds for the explanation of the impulse response function 7**

**from lines 179 to 195; this is a fairly standard mathematical technique and does not need to be explained in so much detail.**

We have removed this lengthy explanation, which we agree is too verbose.

**An important point in the methods chapter is line 220 to 222, where it is stated that the relation between CantML(t) andpCO2ocn contains the effects of changing buffer factor and of changing temperature. Here would be a good place to discuss how the decomposition of effects made here is related to the more traditional feedback analysis that results on the two feedback factors $\beta$ and $\gamma$. I must admit that I wondered why it is necessary to have fitted relation between the two (Appendix B), rather than using standard carbonate chemistry and calculating one from the other, assuming constant alkalinity**

We agree that this needs to be better linked to the introduction. Quantifying carbon climate feedbacks requires a different set of simulations, but their effects are included in our analysis. We have added a discussion of this to discussion section 4.2:

"While uncertainty in the mean state of ocean circulation is most important over the next 60 years, as warming increases, the magnitude of climate-carbon feedbacks related to ocean circulation increase. For simulations made with the one-dimensional model and the CESM, we simulate the effects of real-world climate-carbon feedbacks. The strength of ocean climate-carbon feedbacks (o) in CESM is weaker (-2.4 Pg C $K^{-1}$) than the CMIP5 multi-model mean (-7.8 Pg C $K^{-1}$) (Arora et al., 2013; Friedlingstein et al. 2015). Compared to Cchem, the decline in ocean carbon uptake due to climate-carbon feedbacks in high emission scenarios is an order of magnitude smaller. The one-dimensional model only simulates the reduced CO2 solubility feedback, but is close to the CESM response to warming (Figure 2d), thus indicating solubility effects dominate climate carbon feedbacks prior to 2080. The remainder of

the climate-carbon feedback is related to changes in ocean circulation. In simulations out the year 2300 with CESM (Randerson et al. 2015), or simulations with models featuring a rapidly declining AMOC (Sarmiento et al., 1996), AMOC collapse plays a large role in reducing carbon uptake. The small effect of changing ocean circulation in our simulation is likely because AMOC has yet to collapse by 2080 (Randerson et al. 2015). While assuming that climate-carbon feedbacks related to ocean circulation are small prior to 2080 is consistent with the behavior of the CESM (Randerson et al. 2015), this may not be hold true for the Earth System itself. The uncertainties associated with the timing and magnitude of climate-carbon feedbacks can be avoided by mitigating climate change (Randerson et al. 2015)."

We have added the following justification for the fitted solution to Appendix B:

"There are two main reasons for using a fitted solution. First is that if every variable other than temperature and carbon are held constant, using the full carbonate system equations provides no additional accuracy, thus the additional model complexity is unjustified. Second, we want to be consistent with other models: the concentration scenarios used in CMIP5 (RCP4.5, RCP8.5) are generated using the same model and thus the same representation of ocean chemistry."

**Line 314: "Assuming ocean circulation remains constant" To what extent is that assumption justified?**

We justify this assumption in an updated discussion section 4.2. This updated text is in the response to your comment that starts with "An important point in the methods chapter is line 220 to 222..."

**I do not completely understand, how the fields shown in Figure 4 have been calculated. Is this the zonal average of the 3-d model output minus the**

**expected profile from the 1-dimensional model? How does advection enter this picture?**

We have updated our description of Equation 6, which was understandably hard to interpret. Your interpretation is close, except that the calculation of the expected Cant is derived from the CESM zonal mean Cant in 1990. Three dimensional advection is thus accounted for.

**Section 4.1 contains very little discussion and is in large parts a repetition of results. Then, in the end (lines 468-471), some other results on future Cant uptake are cited, but without giving any connection to what this study has shown. Is there a relation between these results, or is it just inserted here without too much meaning?**

Section 4.1 has been revised for clarity and brevity. The cited works provide the additional context of what is going on with the land sink during rapid mitigation, and what happens to the ocean sink beyond 2080.

**"We find . . .": Is this really a finding, or not rather an assumption that went into the methodology?**

We agree that the current wording is imprecise. Theory predicts that the vertical Cant profile would behave as a function of pCO2atm under exponential forcing but we show that this is true in a complex model. We have updated this line to be more precise:

"We find that an exponentially increasing pCO2atm allows for the vertical Cant profile to behave as a function of pCO2atm, as predicted by the historical scaling."

**Appendix A again explains the impulse response function, even repeats**

**equation 7, and gives details about using it that can be found in many textbooks. I suggest to remove it (including Figures A1 and A2), or shorten it to not more than one paragraph.**

We agree and have done as you suggested, shortening it to a paragraph.

**Appendix B: The coefficients shown in equation B3 to B7 all have different units, but these are not shown.**

Thank you for noticing this omission, units have been added.

**Lines 297 and 298: should one of the mentioned scenarios be RCP8.5 rather than RCP4.5?**

We removed one of the sentences, it was repetitive.

**References: Many references give two web addresses for the cited papers, one being the http-form of the doi (I would rather have the doi without the https://doi.org in front), and the journal address. Are really both necessary?**

We have the same preference but we are using the Biogeoscience's LaTeX template which automatically formats the references.

---

## Author Comment (AC3) · 18 Sep 2020

**Response to Reviewer #2**

Thank you for your detailed review of our manuscript. Following your suggestions, we have carefully edited our manuscript for both clarity and conciseness. Variable names for the various forms of $C_{ant}$ have been changed for clarity, and the introduction and methods have been rewritten. Another large change is the removal of the detailed description of the IRF model in the methods and Appendix. Please consider our detailed responses below:

**1) Structure and conciseness of the manuscript (mainly introduction and**

**methods):**

**\*The concept of an exponential growth of emissions leading to a constant sink rate (under assumptions) is quite central in this work, and it needs to be introduced and put into context. Currently this concept is first mentioned in passing in line 44. It needs to be introduced to the reader before the sink rate is mentioned.**

Thank you for this suggestion, we will introduce this earlier in the manuscript:

"Exponential growth of CO2 emissions leads to a declining sink rate as a result of climate change and reduced chemical capacity of the ocean. Slower than exponential CO2 emissions results in atmospheric growth rate also driving a decline in sink rate."

**\*line 50-53: " Nearly every nation..." I don't see that this sentence adds anything new here, consider deleting.**

Agreed, we have removed this line.

**\*The authors claim that "In the RCP8.5 scenario (Meinshausen et al., 2011), pCO2_atm increases exponentially..." (line 71). To me it is unclear to which degree the RCP8.5 emissions or concentraions can be approximated by an exponential, but this is not very relevant to this study either (since the baseline is an idealized exponential growth). RCP8.5 is the outcome of an advanced modelling exercise, so the emissions are not strictly exponential.**

We have decided to keep this in the manuscript. This helps the reader understand that the behavior of the sinks under RCP8.5 is similar to idealized scenarios with exponential emissions and exponential $pCO_2^{atm}$ increase. Under the nearly exponential

$pCO_2^{atm}$ of RCP8.5, declines in $k_m$ are dominated by changing ocean circulation and changing buffer capacity. In the other scenarios, changes to the atmospheric growth rate play a role in the decline of $k_m$.

**\*Throughout the manuscript, the authors mention exponential historical emissions. It should be made clear that this is an idealization (e.g. by saying "roughly exponential" or similar). This is already the case in some places but missing in others.**

We agree that this makes things unclear and we will update the manuscript accordingly.

**\*lines 54-67: Again, all this could be much more concise. The feedback studies mentioned are considering a single (exponential) concentration pathway, so they cannot quantify an uptake efficiency for different emission pathways. (And yes, in addition to this, they cannot quantify the contribution of a changing buffer factor, either).**

We understand that to someone familiar with climate-carbon feedbacks and the ocean carbon cycle, these lines may be unnecessary, however will keep these lines in the manuscript given the broader readership of Biogeosciences.

**\*line 74-78: Consider moving this up to lines 40-50 where k_s in general is introduced.**

We agree and have moved it up to where $k_S$ is introduced.

**\*lines 102-106: This text is not necessary, we do not need a summary of subsections at the beginning of a section. Please consider removing. The same**

**applies for lines 271-277.**

After further review, we agree and have removed the summaries.

**\*line 108: What is the first sentence of 2.1 supposed to tell us? Just start with "The efficiency metric (eta) used here is defined as k_m ..."**

We agree, and we now begin that line as you suggested

**\*The text explaining equation 4 (lines 112-118) can be shortened substantially: "The historical scaling for ocean C_ant uptake (F_ant) is defined as: (equation 4). The overset "\*" indicates that a variable has been extrapolated using the historical scaling. Here, we diagnose F_ant(1990) from the CESM large ensemble simulations. For example...".**

With another look we agree, and have shortened it as suggested.

**\*Delete unnecessary words, e.g. "mathematically".**

After careful review we have removed many unnecessary words, including "mathematically".

**\*Lines 129-130: "While k_M remains constant,...". This does not reflect the logic of this manuscript. The authors use the exponential scaling to define a baseline against which simulated quantities are compared. The actual k_M is not and does not need to be constant for this exercise.**

As you have noticed this line is misleading and is clearly out of place. We have updated the manuscript to the following:

"We apply the historical scaling to $C_{ant}$ concentration..."

**\*Line 144: "The CESM provides a realistic simulation of the response of the ocean carbon cycle to climate change." What do the authors mean by "realistic"? I would suggest to delete this sentence.**

We have removed the sentence following your suggestion.

**\*Section 2.3: Impulse response functions are a well established tool in climate modelling. It is useful to give a short explanation for those readers that are not familiar and highlight those aspects that relevant for this study, but otherwise the authors should refer the reader to the literature and shorten section 2.3 substantially. Likewise the Appendix A is not necessary. The most important assumption of IRFs is constant circulation. The most important aspect of the Joos-IRF (hidden in the Appendix A) is the fact that, contrary to atmospheric IRFs, the mixed layer IRF take ocean carbon chemistry (including changes in SST) into account.**

We have greatly shortened this section and removed the appendix as you suggested. The response to SST is included in the main text. The temperature variable in the pCO2 equation in the appendix is initial global mean SST. We have updated the text so that there is a separate variable for this temperature ($T_{pi}$)

**\*Throughout section 2, it is unclear what the symbol C_ant(t) is supposed to denote. In equation 7 it is the anthropogenic carbon content of the mixed layer, but otherwise C_ant (often) seems to denote the total ocean anthropogenic carbon. The authors also frequently use the expression "C_ant air-sea flux", which I would suggest to replace by "air-sea flux of anthropogenic carbon" (and**

**use F_ant as a short form of this if necessary).**

We understand that the current usage of variables is very confusing, and appreciate your suggestions. We have updated the ambiguous $C_{ant}$ variables to the following:

| Name | Symbol | Units |
|---|---|---|
| Time, depth, and space dependent anthropogenic carbon concentration | $C_{ant}(x, y, z, t)$ | mmol m$^{-3}$ |
| Mixed layer $C_{ant}$ concentration | $C_{ant}^{ML}(t)$ | mmol m$^{-3}$ |
| Atmospheric anthropogenic carbon inventory | $C_{ant}^{ATM}(t)$ | Pg C |

**\*Section 2.3: The remaining description of the 1d-model is verbose and confusing. Apparently, the authors "extend" the mixed layer IRF downward by "plugging" a diffusion equation under the mixed layer IRF? Yes, the downward flux can be determined "by residual", but then how are the profiles of C_ant(z) calculated?**

These lines should have been removed before submission given that we do not show the downward flux in this version of the manuscript or the $C_{ant}(z)$ from the 1D model. $C_{ant}(z)$ is calculated from the CESM. The 1D diffusion representation of ocean physics for conceptual purposes.

**\*Section 2.5: Equation 12 can be derived by assuming F_ant = F_ant(pCO2atm(t),pCO2ocn(t)). Then, later, it is additionally assumed pCO2ocn=pCO2ocn(C_ant,T). This should be made clearer. (Again C_ant here means surface C_ant).**

We would like to make this more clear, but from this comment it is not clear

what you would like clarified. Hopefully changing $C_{ant}$ to $C_{ant}^{ML}(t)$ makes things more clear.

**\*Section 2.5: The equation del F_ant/del pCO2atm = del F_ant/del pCO2ocn, is this based on Equation 10? Then a minus sigh is missing. Also, the dependency of the transfer velocity on temperature is neglected in this step. How does it follow from Equation 12 that "The pCO2ocn closely follows pCO2atm, and the sign of their growth rates is the same"?**

Thank you for catching the missing minus sign. In our simplified model the transfer velocity is independent of temperature. Also, the statement "The pCO2ocn closely follows..." is not derived from the equation 12. It's a useful heuristic for understanding the equation that is derived from the behavior of the model.

**\*Section 3.1: If a paragraph begins with "In the RCP4.5 scenario, changes to the spatial pattern lie somewhere between RCP8.5 and the 1.5C scenario" both scenarios should have been discussed already. This is not the case here**

Thank you for catching this mistake, we no longer reference RCP8.5 before discussing it.

**2) In my opinion, the term "historical scaling" used by the authors is misleading or at best confusing. The sink rate has not been constant over a substantial part of the historical period in observations (Raupach et al. 2014, cited) as well as in the model experiments used in this study (Fig S1). If the authors wish to replace the previously used term "transient steady state", why not just saying what it is, e.g. "exponential scaling" (or something similar that does not refer to the historical period)?**

Thank you for your feedback, we see how this is confusing in the current version of the manuscript so we have updated the introduction. Exponential CO2 emissions growth is a necessary, but not sufficient condition to ensure that $F_{ant} = *F_{ant}$. The historical scaling of $F_{ant}$ holds if the following conditions are met: the impacts of climate change are small, ocean chemistry is relatively unchanged, and emissions continue at a exponential rate. Over the historical period, changes in these conditions are small, therefore $F_{ant} \approx *F_{ant}$. In Figure 3a from Devries (2014), it is evident that observational estimates of the increase in $F_{ant}$, is nearly proportional to the increase in $pCO_2^{atm}$ , therefore we can assert that variability in $k_M$ doesn't make the long term change in $F_{ant}$ inconsistent with the historical scaling. We refer to the scaling as the historical scaling because the necessary conditions are only satisfied over the historical period, in the RCPs these conditions are not all satisfied.

**4) Choice of time periods:**

**The time period 1920-2006 is not the "historical period". From a CMIP5 perspective this would be 1850-2006. Why do the authors choose 1920 as a starting year? Likewise, why do the authors not use the last 20 years of the scenarios, which would be most interesting period in the mitigation scenarios?**

The choice of time period was set by the length of NCAR's simulations. The historical period of the CESM ensembles begins in 1920, which differs from the CMIP5 protocol (1850 starting year). The CESM ensemble for RCP4.5 ends in 2080, while the RCP8.5 and 1.5C scenarios end in 2100. From 2080-2100 in the 1.5C scenario, the air-sea flux remains close to 0, thus 2080 is a natural cut off.

**5) 1d-model evaluation:**

**To me it is not given that the 1d diffusion model has skill in reproducing**

**the CESM global mean C_ant profiles for all scenarios, but this is the basis of the analysis of the "gradient effect".**

We will update our manuscript to emphasize that our diagnosis of the gradient effect is not based on the 1D diffusion model. Changes to downward anthropogenic carbon transport are either due to changes in the circulation or the gradient of $C_{ant}$. In experiments with ocean GCMs, changes to downward anthropogenic carbon transport due to changing ocean circulation are small (Winton et al. 2013, Bronselaer and Zana 2020). Thus, regardless of whether the circulation is parameterized as diffusive as in the 1D model, or a mix of diffusive and advective processes as in the CESM, the change in vertical transport is largely due to changes in the vertical gradient of $C_{ant}$. In conclusion, we can diagnose the gradient effect directly from the CESM simulations.

**The space saved by shortening Section 2 could be invested in presenting a brief evaluation of the (full) 1d-model compared to CESM. How well does the fitted 1d-model reproduce F_ant in the 3 different scenarios?**

The 1D-model's representation of $F_{ant}$ is shown in Figure 2d. $F_{ant}$ as simulated by the 1D model is almost identical to the CESM simulations of $F_{ant}$, as a result of the tuning process.

**More important, how well are vertical profiles of C_ant simulated? To me it is not given that the 1d diffusion model has skill in reproducing the CESM global mean C_ant profiles for all scenarios, but this is the basis of the analysis of the "gradient effect".**

In order to clarify any potential confusion, we have updated the manuscript to make it more clear that the profiles of $C_{ant}(z)$ shown are only from the CESM. In fact, with the pulse response form of the model, we do not simulate the 1D model's vertical

profile.

**6) Section 3.4:**

**This section is not easy to follow. What is the main point here? I guess it is the fact that the ocean uptake in the strong mitigation scenario after 2040 is maintained by the ocean through continuous downward mixing (otherwise the surface ocean would start outgassing because pCO2atm declines already). Could the authors please add some easy to understand explanations here?**

Your interpretation is correct. This section quantifies how the atmospheric growth rate supports the air-sea flux in the RCP4.5 and RCP8.5 scenarios, and acts to decrease the air-sea flux in the 1.5C scenario.

**Further, related to my point 5) above, how realistic is this process simulated by a 1d-model?**

This is the most simplistic way to view the ocean anthropogenic carbon air-sea uptake, but has been shown by many authors to be very useful in the study of the carbon cycle (Joos et al., 2013; Raupach et al., 2014; cited).

**In reality we would have upwelling of waters that have been last in contact with the atmosphere in preindustrial times, that can potentially sustain ocean uptake even under declining CO2, but this is not the case in the 1d-diffusive model. Here the processes must be different. Could the authors please comment on this?**

Advection and diffusion both act to mix anthropogenic carbon downwards in the ocean. Although the upward and downward advective fluxes are not necessarily

colocated (e.g. Southern Ocean upwelling, subtropical downwelling), we are only interested in the integrated effect of advection on the global air-sea flux. Thus, the vertical mixing of anthropogenic carbon can be conceptualized as a 1D diffusion process.

Secondly, the HILDA model, which the mixed layer response function is derived from, includes a representation of advection and diffusion. As the impacts of climate change on ocean circulation increase, the advective and diffusive processes respond differently; however, over the next 100 years changes in uptake related to these transport processes are small (Winton et al., 2013, Bronselaer and Zana 2020).

**References**

Bronselaer, B., Zanna, L. Heat and carbon coupling reveals ocean warming due to circulation changes. Nature 584, 227–233 (2020). doi:10.1038/s41586-020-2573-5

DeVries, T. The oceanic anthropogenic CO2 sink: Storage, air‐sea fluxes, and transports over the industrial era. Global Biogeochem. Cycles 28, 631– 647 (2014), doi:10.1002/2013GB004739.

Joos, F., Roth, R., Fuglestvedt, J. S., Peters, et al. Carbon dioxide and climate impulse response functions for the computation of greenhouse gas metrics: a multi-model analysis. Atmos. Chem. Phys. 13, 2793–2825 (2013), doi:10.5194/acp-13-2793-2013

Winton, M., S. M. Griffies, B. L. Samuels, J. L. Sarmiento, and T. L. Frölicher Connecting Changing Ocean Circulation with Changing Climate. J. Climate, 26, 2268–2278 (2020), doi:10.1175/JCLI-D-12-00296.1

---

## Author Comment (AC4) · 18 Sep 2020

Your comment is greatly appreciated. We agree that it should be made more clear that the IRF model is a simple parameterization of the complex processes that transport anthropogenic carbon into the ocean interior. We have updated the methods, as well as the introduction and conclusion, to emphasize the role of processes such as mode water formation and overturning (Iudicone et al., 2016) in the vertical transport of anthropogenic carbon. In our updated manuscript, we highlight the important recent advances in process understanding, and clarify the differences between our model's representation of ocean transport and real world processes.

---

## Author Response (AR2)

**Columbia University**

IN THE CITY OF NEW YORK

LAMONT-DOHERTY EARTH OBSERVATORY

March 18, 2021

Dear Associate Editor Fortunat Joos –

Thank you for your comment and those of the Reviewers. They have helped us to substantially improve this manuscript.

In addition to the detailed responses to all three sets of comments that are appended, we wish to highlight several key changes made with this Revision:

1. The need to clarify the impact of warming was highlighted in the review, and this led us to explicitly separate its effects in our IRF decomposition (Table 1, Figure 5). The effects remain small, as originally stated, now they are explicitly quantified and clearly presented.

2. In reviewing IRF scripts so that Figure 5 could be remade, we identified a unit conversion problem, and this resulted in the need to retune the IRF model to match the CESM. We now use h=51m to replicate CESM. This leads to ocean carbon uptake reductions in RCP8.5 and RCP4.5 both being dominated by the impact of carbonate chemistry. Now, only the 1.5C scenario has significant slowing attributable to reduced transport of carbon from surface to depth --- this transport change is attributable to a large reduction of the global-mean vertical gradient of Cant from surface to depth in the ocean (Figure 3c, Figure 4), given the assumed constant ocean circulation.

3.  We have removed all mention of "one-dimensional model" and replaced this terminology with IRF. In addition, we have explicitly stated in several places that the decomposition equation including Kz is meant to be conceptual and to assist in understanding decomposition approach, but does not represent the IRF itself.

4. We have made substantial revisions to the text throughout to integrate the updated results and to clarify the presentation.

Thank you again for your careful attention to this manuscript,
Galen McKinley

Sincerely,

Galen McKinley, Professor, mckinley@ldeo.columbia.edu
Sean Ridge

P.O. Box 1000     61 Route 9W     Palisades, NY  10964-8000  USA     845-329-2900     http://www.ldeo.columbia.edu

**Associate Editor Decision: Publish subject to minor revisions (review by editor)** (02 Mar 2021)
by Fortunat Joos
Comments to the Author:
Dear authors

Your revised manuscript has been evaluated by the two reviewers. Both find your manuscript a useful contribution to the field and suggest publication after minor or technical revision. Based on their recommendations and my reading, I ask you to further revise the manuscript before a potential publication.

My own comments and suggestion are added below. Most notably, I find the terminology regarding the "gradient effect" misleading. The effect termed "gradient effect" arises, in my understanding, from the time-varying evolution in atm. $CO_2$ and its deviation from an exponential curve. This should be reflected in the name for this effect. Section 2.3 is, in my opinion, not very clear and requires further improvement. I also suggest renaming "one dimensional model" to Impulse Response Function (IRF) model throughout the manuscript. As noted by one of the reviewers, there is some confusion regarding the representation of gradients in the IRF model and how this is described in the manuscript.

Thank you for submitting your work to Biogeosciences. I am looking forward receiving your revised version.

Yours sincerely,
Fortunat Joos

**Major Comments from the Associate Editor**

1) The terminology of the attribution of Canth to mechanisms is confusing and misleading: The CESM and IRF model results for Canth are compared to a historical scaling approach. The deviations of ocean uptake are attributed to "carbonate chemistry" and "a vertical gradient effect" (Eq. 11).

The historical scaling is taken as a reference to discuss mechanisms that lead to deviations from these scaling results. The assumptions of the historical scaling to work are (i) the system is linear, i.e. no change in carbonate chemistry and circulation, and (ii) the forcing of the system is exponential, i.e. the atmospheric $CO_2$ follows an exponential curve. Given this, it seems natural to attribute deviations to (i) carbonate chemistry (as done by the authors), and (ii) to deviations from exponential forcing – an atmospheric $pCO_2$ history effect.

The deviation in the vertical gradient between Canth estimated from historical scaling and simulated by CESM (Fig. 3 and 4) arises both from the carbonate chemistry (changes in buffer factor) as well as due to deviations of the atmospheric $pCO_2$ evolution from an exponentially increasing curve. (In addition, there may be changes in circulation, wind speed and air sea gas transfer coefficient and in the marine biological cycle that also may affect the simulated vertical

gradient in Canth in CESM – albeit these effects are likely of minor importance here as postulated by the authors).

We concur that the historical scaling is based on these assumptions, and that it is already established the historical scaling will not hold as the atmospheric becomes sub-exponential (Raupach et al. 2014). This is described in the paragraph immediately after equation 2.

Our goal is to better understand what are the ocean physical / chemical processes that will cause the deviation from historical scaling. CESM does not allow a separation of the chemical and the physical effects, and so we use the IRF model to estimate this separation.

The historical scaling is a useful reference point because it has been very useful in understanding the carbon sink to date, and analyzing observed interior Cant accumulation (Gruber et al. 2019, Science). But it will not hold going forward, and it is important to understand that the mechanisms by which the sink will diverge will depend on emissions.

The problematic notation is evident when comparing section 3.3 and 3.2. On line 300 it is stated: "These two effects reduce the ocean carbon sink by a total of 31% from the historical scaling, with approximately 1/3 of the effect due to the vertical gradient and 2/3 due to carbonate chemistry." In section 3.2 and Fig. 3 the deviation in the Canth gradient between the historical scaling and as simulated by CESM is shown. It is clear that the integral of these deviations (shown by gray shading) must equal the difference in cumulative air-sea flux of Canth for the historical scaling and for CESM. Thus, about 2/3 of the difference between the vertical gradient in Canth as simulated by CESM and the historical scaling is due to ocean chemistry and not due to the "vertical gradient effect".

What is termed "vertical gradient" effect in this paper, reflects, in my opinion, rather the effect of a time-varying CO2 evolution which deviates from an exponential curve. Thus, I find it misleading and conceptually confusing to talk about a "gradient effect". The terminology should be changed.

What we have been calling the "vertical gradient" effect is not the deviation of the atmosphere from its exponential curve. Instead, it is the excess Cant stored in the thermocline that modifies the near-surface gradient of Cant, and that raises surface ocean pCO2 via re-entrainment / reemergence from below. It is a primary goal of this work to illustrate that as the atmosphere pCO2 growth rate slows or becomes negative, this effect will be increasingly important to setting the magnitude of the ocean carbon sink.

To make our focus on the physical transport of carbon more clear, and because we are focused mostly on the global-mean profile, we have renamed this "vertical transport of Cant" throughout the manuscript.

The profiles in Figure 3 and 4 are for the full CESM, including both the effect of carbon chemistry and of ocean vertical transport of Cant. These figures illustrate how the vertical

profile differs from the historical scaling under each emission scenario. With Figure 5, we use the IRF model to attribute how much of the change in these profiles is caused by carbon chemistry as opposed to by the ocean vertical transport of Cant.

We have carefully reviewed the text and made an effort to clarify these points throughout.

2) Please replace the term "one dimensional model" with "Impulse Response Function model " and "IRF model". Reason: The IRF model is not a 1-d model and may represent 3-d, 2-d, as well as 1-d models. The IRF model does not resolve a 1-d, e.g. vertical, gradient.
Thank you. We have made this change, and it has much clarified the manuscript.

3) Assumptions for the diagnostic framework and the IRF should be clearly stated.
a) It seems that you apply the response function from the HILDA model and adjust mixed layer depth to emulate the anthropogenic carbon uptake by CESM. Thereby it is assumed that the time scales of anthropogenic carbon removal from the mixed layer and thus of ocean overturning are the same (or at least scale in some sense) in the HILDA model and in CESM-POP2. The IRF model does nicely emulate CESM over the 21st century. However, I doubt that this is true on longer time scale as the ventilation of the deep Pacific and Indian is much too sluggish in POP2 as evident from radiocarbon simulations. It should be noted that the IRF model may not equally well represent CESM results on longer time scales.
This is correct. We use the HILDA response function to emulate CESM. We state this more clearly in the last paragraph of section 2.3. We add a last sentence here " It is important to note that despite the ability of the IRF model to emulate CESM behavior through 2080, this does not mean it would be able to emulate longer timescales; particularly under high emissions, greater ocean circulation and biogeochemical changes are expected beyond 2100 (Randerson et al., 2015)."

b) In section 2.5, it is assumed that the gas exchange velocity k_gas is time invariant. Again this may not hold under future climate change in CESM and in reality. Similarly, changes in the marine biological cycle are neglected.
Yes, this is correct. To make this assumption more clear, we now start this section (now section 2.4) with "Considering anthropogenic perturbations on top of a background natural state, the air-sea flux of anthropogenic carbon is a function of the pCO2 in the atmosphere and ocean (Equation 8), and pCO2 is a function of the anthropogenic carbon content (Cant) and the temperature (T):  Fant(pCO_2^atm, pCO_2^ocn(C_ant, T)).  Change gas-exchange rates are assumed negligible, and because the biological pump is part of the background natural cycle, it is also assumed constant."

c) ocean circulation is assumed to be time invariant and not affected by global warming. This is stated. However, it also applies to Eq. 14 where the last term only reflects the impact of warming on carbon chemistry, but not on ocean transport and stratification.
This is correct, we have added note of this.  We have also explicitly separated out the impact of warming  on solubility for additional clarity.

4) There is technically nothing wrong with section 3.4 and and Fig. 6. However, I do not see that this discussion and analysis adds any particular insight. I suggest to delete this text and figure from the MS. It is well established that the surface ocean pCO2 equilibrates quickly with the atmospheric pCO2 (typical time scale of 1 yr, see e.g. Broecker and Peng, Tracers in the Sea, 1984) and that the air-sea disequilibrium in pCO2 is small relative to the absolute pCO2 value. Correspondingly, the gross air-to-sea and the gross sea-to-air CO2 flux are very comparable. This fact needs not to be iterated.

We have removed this section as suggested.

Further Comments

Line 17: I find the number of 39% misleading. Ocean uptake should be related to total anthropogenic emissions, including those from land use. LUC emissions are substantial over the historical period. A more conventional estimate is that the ocean has taken up between 25 to 30 percent of anthropogenic emissions.

We respectfully disagree on this point. It is our carefully-considered conclusion that as long as the denominator is clearly cited as being fossil and cement emissions, as is the case here, it is not inaccurate to cite this number. The value is taken directly from the Global Carbon Budget 2019 (their Figure 9), and the same choice of denominator has been used and presented very prominently – i.e in the abstract - of Sabine et al. 2004 (Science). We recognize that many authors choose to include the Land Use source in the denominator to arrive at "total anthropogenic emissions", but cumulatively, the land was a net source until ~1950 and only after that began to trend to a sink. Moreover, the Land Use component on its own is very poorly quantified. On the other hand, from ocean data (Sabine et al. 2004, Gruber et al. 2019), we have a good estimate of what the cumulative uptake by the ocean of excess carbon has been, and from this we can get a relatively-low-uncertainty estimate of the net land flux, as demonstrated by Khatiwala et al. 2009 and replicated in Figure 1 of McKinley et al. 2017 (Annual Rev. Marine Science). We see no reason to incorporate the very uncertain land-use source into the denominator, and would like to maintain consistency here with our group's approach by citing the magnitude of the cumulative ocean sink relative to fossil fuel and cement emissions. To make it more explicitly clear that we are talking about "excess carbon in the ocean", which is what the studies of Sabine 2004 and Gruber 2019 are able to quantify, we have revised the sentence. It now reads "The ocean has absorbed excess carbon equivalent to 39% of the $CO_2$ from industrial era fossil fuel combustion and cement production (Friedlingstein et al., 2019). "

L30: effective surface diffusivity: The term "surface" seems misleading. It is a vertical diffusivity applied to operate over the upper and deep ocean

We have removed the mention of diffusivity from this sentence.

l. 38: natural sink – suggest deleting "natural"

Thank you. We have made this change.

l40: expand to say: "are strictly exponential and the system is linear". As this holds only for linear systems.

Thank you. We have made this change.

L124: historical period: 1920 to 2006: Should this not read 1820 to 2006? Starting the simulation in 1920 would lead to a massive cold start problem. I think the RCP simulations started in 1820.

Thank you. This now reads "Following a long preindustrial spin-up, all simulations used here are forced for the historical period (1850-2005) with observations of pCO$^{atm}$. The individual ensemble members of the Large Ensemble are branched off at 1920 (Kay et al., 2015)."

Section 2.3: I found this section somewhat unclear and confusing.

It is not clear to me what is done. I guess that you use the IRF function of HILDA from Joos et al. and adjust the mixed layer depth h to 109 m to match the anthropogenic carbon uptake as simulated by CESM for the historical period. Eq. 7 and Eq. 10 (together with the chemistry described in the appendix) are then used to compute ocean uptake. If this is correct then please state this explicitly. It would also help to bring Eq. 10 next to Eq. 7.

It is also my interpretation that Eq. 8 and 9 are not applied in this study, but added for illustration and to conceptually separate mechanisms. I recommend moving these two equations and the related text to section 2.5 to keep the description of the IRF model strictly separate from the description of the separation into individual terms.

Thank you. We have removed these equations and streamlined section 2.3 to clarify the IRF model.

L145: typo: in -> on

Thank you. We have made this change.

L145: Change "one dimensional" to "reduced form" or to "substitute".

Thank you. We have changed this to "reduced form" throughout the manuscript.

L160ff, eq. 8: Eq. 8s represents rather a conceptual than an operational description and the text on l.160 is in my opinion not correct. Even in the case of 1-d box diffusion model, many ocean layers are used to determine the carbon flux at the base of the mixed layer. Here it is not clear how the gradient dC_ant^ML/dz would be derived. I suggest rewriting the text to read something like "The convolution integral (Eq. 7) may be conceptually linked to the following tendency equation of anthropogenic carbon in the surface ocean mixed layer:
Eq. 8
The change in the mixed layer carbon concentration results from the air-sea flux (F_anth) and from the flux between the mixed-layer and the deeper ocean, here formally written as a diffusive flux.

Thank you. We have fully changed this discussion, and moved it to section 2.4.

Suggest deleting "The one-dimensional model'sKz;eff must match that of the ocean component of the ESM being emulated (Gnanadesikan et al., 2015)." as this sentence is not needed and rather confusing given that it remains unclear how the gradient is determined.

Thank you. We have made this change.

L 167: Please specify from what the "diffusive flux term" is estimated. I first guessed from the air-sea flux and the change in surface layer concentration as simulated by CESM. However, then in section 2.4 it becomes clear that the separation is done using the IRF model.

Thank you. We have substantially clarified these sections and this test no longer appears.

L198: only true under the assumption of a constant k_gas and in a spatially-averaged framework. These assumptions may not necessarily hold for CESM. Please clarify the limitation.

Thank you. We have fully changed this discussion; moved it to section 2.4.  We clarify that the

use of Kz and the whole equation 12 is meant to be conceptual only.

Line 207: expand to read: "impact of warming on the carbon chemistry"

Thank you. We have made this change.

Line 259: typo: fro -> from

Thank you, correction made.

Section 3.2: The deviations in Canth between the historical scaling and POP2 are due to changes in the buffer factor and due to deviations from an exponential forcing in the scenarios. Correct? I suggest to state this for clarity and to avoid confusion with the attribution done in the next sections.

We believe that our clarification in terminology to "ocean vertical transport of Cant" has resolved this issue.

Line 318: two verbs: "acts increases"

Thank you, correction made.

L 324: typo: around

Thank you, correction made.

L385 to 391: This paragraph is misleading and should be deleted. An IRF model is applied and not a 1-d diffusion-advection model.

We have substantially modified this paragraph to correctly state that the IRF model emulates and array of physical processes in the CESM.

L457 ff: Suggest to use the term Impulse Response Function model instead of 1-d model throughout the MS.

Thank you, we have made this change throughout.

**Review 1** of the first revised version of "Ocean Carbon Uptake Under Aggressive Emission Mitigation" by Sean Ridge and Galen McKinley, submitted to Biogeosciences Discussions https://doi.org/10.5194/bg-2020-254

General Comments

The revised manuscript by Ridge and Galen takes into account most of the critical points that another reviwer and I have raised in the first version. It is much more concise now, much clearer in its nomenclature, and also puts the results more into relation with previous work.

As a result, the manuscript has become quite helpful for understanding how future uptake of carbon in the ocean will evolve at least in one particular earth system model. The methodology proposed here may also be useful to analyze the results from other models. I think it can now be accepted for publication in Biogeosciences.

Thank you for these detailed comments. We have addressed them, as noted below.

Line 108: empty space missing after $C_{ant}^{ML}$.

Thank you, correction made.

Line 198: Isn't there a minus sign missing in the inlined equation?

Thank you. Yes, this was incorrect, but it has been removed in the revision to section 2.4.

Line 259: 'fro' should be 'from'

Thank you, correction made.

Line 289: Capitalise the 'V' in DeVries, 2014

Thank you, correction made.

Line 307: the degree in missing in 1.5C

Thank you, correction made.

Line 324: 'aroudn' should be 'around'

Thank you, correction made.

Last line, figure caption 6: Missing space after $C_{ant}$

This section has been removed per the recommendation of the associate editor.

Line 355: The sentence beginning with 'If emissions' is inconsequent in its usage of a 'the stronger $\ldots$ the larger' construction.

This has been revised to read "As emissions are mitigated, the back-pressure effect grows (Figure 3-5)."

Line 361-362: I don't understand this sentence.

This sentence has been removed as part of an overall effort to clarify this paragraph.

Line 368: Too much space after $C_{ant}$

Thank you, correction made.

Line 400: 'Magnitude' should be plural, I think

Thank you, correction made.

References:

At least in Friedlingstein et al, 2019, and in Sanderson et al., 2017, the http-form of the doi contains errors.

CO$_2$ is written without subscripting the 2 in Khatiwala et al.,2009, Peters et al., 2017, Tanhua et al., 2007.

In Sanderson et al., 2017 a special character has sneaked into the ASCII text

In Takahashi et al., 1993, the journal name is incomplete.

Thank you, these corrections to the references have all been made.

**Review 2** of a revised version of "Ocean Carbon Uptake Under Aggressive Emission Mitigation" by Sean Ridge and Galen McKinley

I have reviewed a first version of this manuscript, and I am happy to see that the revised manuscript has improved a lot. I have a few remaining issues (listed below), and I would recommend the manuscript for publication in Biogeosciences after the authors have addressed these.

Thank you for these detailed comments that have helped us to improve the presentation of our work. We address all the comments, as indicated below.

Main point:
* * *
The analogue of vertical diffusion used by the authors is misleading as far as the impulse response function is concerned. In lines 160-168 it is stated: "The convolution integral (Equation 7) is derived from the model's surface anthropogenic carbon tendency equation", which is not correct. The impulse response function (IRF) doesn't know the gradient of anthropogenic carbon, and I don't see how equation 9 and 7 are related. Of course, it is possible to reproduce results from a box-diffusion model sufficiently well by fitting an IRF (as in Joos et al. 1996). But an IRF is not based in any way on the assumption that the vertical mean ocean state can be represented by a 1d-diffusion approach as in equation 8. The IRF is an empirical fit to model results, representing all processes that the fitted model included.

Thank you for your helpful comment. We agree that this was not clear. We have removed the terminology "one-dimensional model", replaced it with IRF throughout. We have made it clear that the discussion of Kz is only a conceptual component, not an actual fitting in section 2.5. We have made the most changes in section 2.3, and have made substantial terminology changes also in the abstract and discussion.

Later we find statements like

*lines 385-388: "Our one-dimensional ocean carbon cycle model represents multiple physical processes that remove carbon to depth as a single diffusive process that is constant in time (Equation 8) using an effective vertical diffusivity, K_z,eff . The value for this term in the one-dimensional model has been set (Section 2.3) so as to mimic advective, eddy-diffusive and watermass transformation processes occurring in CESM.", which wrongly suggests that the IRF would use a vertical diffusivity as a parameter.

Thank you for this comment. We have clarified this paragraph, and also in methods, to make it clear that the IRF model does not have Kz,eff as a tuning parameter. The point that is now more clear is that the IRF is emulating advective and diabatic processes.

*lines 406-408: "The remainder of the climate-carbon feedback is related is due to changing physical transport, which in the one-dimensional model is due only to the vertical carbon gradient and ocean circulation is constant." Again, the IRF does not know the vertical gradient

of carbon.

Thank you for this comment. We have clarified that the impact of "ocean vertical transport" has been diagnosed using CESM in this work.

I would strongly suggest that the authors revise the above mentioned parts of the manuscript. I see that the authors need to introduce the vertical C-gradient somehow, to be able to define the "gradient effect", but please do this in a way that avoids the impression that this is parametrized in the IRF.

Thank you for this comment. This is the most important clarification with this round of review.

Also, it would be good to mention/discuss that the "gradient effect" as used by the authors is found by residual (difference between the "historical scaling" and the ccc-simulation) so you don't actually need a model that models a vertical gradient to derive it.

Thank you for this comment. We have clarified that the impact of "ocean vertical redistribution" has been diagnosed using CESM in this work.

Minor points:
* * *
The concept of a constant sink rate, which allows for defining the "historical scaling" is a central point, but it is introduced too late in the text (lines 63-70). It would make the text easier to understand if these lines could be moved up to somewhere after the definition of k_S (lines 34-44). I see that lines 63-70 deal with the ocean sink only while lines 34-44 more generally deal with the land and ocean sink, but then please generalize lines 63-70 (you write about "...the theoretical prediction of constant sink efficiency..." already in line 40, and for a reader not familiar with the concept this is unclear).

Thank you, we have moved this paragraph up as requestred. We have also removed the potentially-confusing, statement "This result is as expected because the theoretical prediction of constant sink efficiency is only valid if CO2 emissions are strictly exponential, and the system is linear." The theory for the ocean will shortly be explained, and this is what is critical for the reader to understand.

line 50-51: Please consider replacing "climate-carbon feedbacks" by "carbon cycle feedbacks" (here both, carbon-climate and carbon-concentration feedbacks are meant).

Thank you, this change has been made.

line 74: RCP8.5 is not a "business-as-usual" scenario, it should be called a "high emission" scenario.

Thank you, this change has been made.

line 100-105: Please double check whether this could be more concise. It seems to me some sentences are more or less duplicate.

Thank you, we have made this section more concise.

Section 2.2: Please spell out which version of CESM you are using and add a reference for the whole model. Please also add a suitable reference for the RCPs (e.g. 10.1007/s10584-011-0148-z). It remains unclear to me why the authors are reluctant to properly acknowledge CESM scientists and engineers, as I commented in my first review.

Thank you. We apologize for this oversight. We now reference Hurrell et al 2013 at the start of section 2.2. To further emphasize, to the acknowledgements, we add "We acknowledge the CESM Large Ensemble Community Project and supercomputing resources provided by NSF/CISL/Yellowstone. This material is based upon work supported by the National Center for Atmospheric Research, which is a major facility sponsored by the National Science Foundation under Cooperative Agreement No. 1852977. We thank all the scientists, software engineers, and administrators who contributed to the development of CESM We thank all the scientists, software engineers, and administrators who contributed to the development of CESM."

lines 177-178: "...we perform two sensitivity experiments...". It is only one sensitivity experiment that is performed, isn't it? The "historical scaling" is not an experiment with the 1d-model. Also, please describe the sensitivity experiment briefly in the main text (not only in Table 1). E.g. "...we perform a sensitivity experiment ... where the buffer factor is kept at pre-industrial level" or similar.

Yes, you are correct that in the last version there was only one sensitivity experiment. Now that we have added the explicit separation of warming, there are two experiments as stated.

line 188: From Figure 6 in Randerson et al. (2015) I roughly read that AMOC is reduced by 10 Sv in 2080. This is quite a significant reduction. So the point is not that CESM hasn't significant circulation changes but rather that the carbon cycle in CESM seems to be relatively insensitive to these changes (on a global scale). Please consider revising this sentence.

We have modified this to "The physical circulation is assumed fixed in the IRF model, consistent with the carbon cycle in CESM not being sensitivity to changes in circulation over 1920-2080 (Randerson et al., 2015)."

line 199-200: "The $pCO2ocn$ closely follows $pCO2atm$ , with the same sign." I don't understand what the authors mean with "with the same sign". Please clarify.

Thank you, we have removed "with the same sign", as it is unnecessary.

line 217: "Projected Spatial Redistribution of the Anthropogenic Carbon Air-Sea Flux". I think "redistribution" is not a good wording here. Maybe better "Projected Spatial Patterns of Anthropogenic Air-Sea Carbon Flux"?

Thank you, we have changed this to read as suggested.

lines 290-291: "...the ocean would absorb 158 Pg C_ant from 2020 to 2080". You mean would absorb 158 Pg C_ant in addition, right? Please consider making this clearer.

This now reads "Due to the fact that ocean chemical capacity changes in the future, uptake is reduced significantly, -233 $PgC_{ant}$ from 2020 to 2080 from what it would be without chemical change (light blue shade). "

line 304-305: Please check the logic of this sentence (it is not clear what "because" refers to).

Thank you, we have clarified this. It now reads "The $\Delta C_{chem}$ effect is the weakest in this scenario, -37 Pg C in 2080. The weak $\Delta C_{chem}$ effect is consistent with this scenario taking up the least anthropogenic 305 carbon because chemical capacity decreases as anthropogenic carbon uptake increases."

line 392: "differences" is a bit unclear. I think the authors mean "changes".

We have revised this to read "For the historical period, global-mean air-sea fluxes and anthropogenic carbon storage are not substantially different across models, despite these models having substantial differences in the ocean circulation …"

lines 400-402: "We use our one-dimensional model to estimate climate-carbon feedbacks for CESM." If this is really the case a bit more explanation would be appropriate, but it seems the authors use the feedback values from Arora et al. (2013). Please add either more explanation as to how you estimate the feedback with the 1d-model, or delete this sentence. Also CESM's gamma_o is not only weaker than the CMIP5 mean, but it has the weakest ocean carbon climate feedback of all models.

We have eliminated this paragraph as it was not central to our discussion.

lines 403-404: "For CESM, decline in ocean carbon uptake due to climate-carbon feedbacks in high emission scenarios is an order of magnitude smaller than due to change in ocean chemistry (Randerson et al., 2015)." I cannot see that the Randerson paper supports this statement. They cannot separate climate-carbon feedbacks from feedbacks due ocean chemistry with their experiments. Please clarify/justify this statement.

We have eliminated this paragraph as it was not central to our discussion.

line 431: Please specify after which year carbon in the upper 100m starts to decrease.

We have revised this to read "… the anthropogenic carbon concentration begins to decrease starting in 2038, just two years after the maximum pCO2atm of 437ppm is achieved."

Technical:
* * *
line 24: "is controlled" -> "is further controlled"

This change has been made.

line 27: "...dominates regional patterns anthropogenic..." -> "...dominate regional patterns of anthropogenic..."

This change has been made.

line 48: "that the future" -> "that in the future"

This change has been made.

line 59: This sentence duplicate the previous one, please consider merging the two sentences.

This has been revised to read "Climate driven effects stem from the warming of the surface ocean, which reduces gas solubility, and ocean circulation, thus reducing the efficiency of ocean uptake (Friedlingstein et al., 2013)."

line 82: Please consider writing "... referenced to the year 1990, and expressed as a percentage".

This change has been made.

line 145: "based in impulse response functions" -> "based on a impulse response function"

This change has been made.

line 147: "...to the RCP4.5 and RCP8.5 CO2 concentration pathways" -> "...to the RCP CO2 concentration pathways" (applies to all RCPs, not only 4.5 and 8.5)

Thank you, this sentence now reads "….and is also used for all RCP scenarios to convert projected emissions to CO2 concentrations (Meinshausen et al., 2011)."

Section 2.5: There is a couple of instances where the superscript "ML" is missing in the notation for "C_ant^ML". Please check this throughout this section.

Thank you, we have checked to make sure that CantML is referenced when it should be. When

Cant in the ocean more broadly is intended, we use Cant.

line 198: There is a minus sign missing in the equation, please check this.

Thank you. This was incorrect, but has been removed in the revision to section 2.4.

line 233: "Globally-mean" -> "Global-mean"

This change has been made.

line 239: "CESM-simulated air-sea anthropogenic carbon uptake" -> "CESM-simulated anthropogenic carbon uptake"

This change has been made.

line 233: "(Figure 2b)" -> "(Figure 2c)"

This change has been made.

line 249: "at that location" -> "at that depth"

This change has been made.

line 259: "fro" -> "from"

This change has been made.

line 276: "by the the" -> "by which the"

Thank you, correction made.

line 318: delete "acts"

This section has been removed per the recommendation of the associate editor.

line 324: "aroudn" -> "around"

This section has been removed per the recommendation of the associate editor.

line 326: "to a less net uptake" -> "to less net uptake"

This section has been removed per the recommendation of the associate editor.

line 362: "...are renewed connection..." Please check grammar of this sentence.

This sentence has been removed to clarify this paragraph.

---

## Author Response (AR3)

**Columbia University**

IN THE CITY OF NEW YORK
LAMONT-DOHERTY EARTH OBSERVATORY

March 25, 2021

Dear Associate Editor Fortunat Joos –

Thank you for noting the need for these additional corrections. We address each one as follows:

1)      Line 2: It is incorrect to say "UNFCC Paris Agreement, committing to mitigate global anthropogenic carbon emissions so as to limit global mean temperature increase to no more than 1.5_C." Please see the Paris Agreement and change the text accordingly.

We have revised this to read "Nearly every nation has signed the UNFCC Paris Agreement, committing to mitigate anthropogenic carbon emissions so as to limit global mean temperature increase to well below $2^{\circ}$C compared to pre-industrial, and to pursue efforts to limit the increase to $1.5^{\circ}$C. "

2)      Line 450: "Cwarm only accounts for the impact of warming on solubility ". In my understanding, this should read here and elsewhere (e.g., in the caption of Fig. 5) in the MS: "Cwarm only account for the impact of warming on solubility and inorganic carbonate chemistry" as chemical equilibrium constants also change with temperature.

Yes, this is correct. We have changed "solubility" to "solubility and inorganic carbonate chemistry" throughout.

3)      Fig. 5: The present coloring is inconsistent with the attribution presented in Tab. 1. Please color the difference between the run with constant chemical capacity (Cccc) and the historical scaling always by light green. Now this difference, denoting the "transport effect" according to Tab. 1 is colored by light blue if the "transport effect" is positive. As evident in Fig. 5, the "transport effect" as determined with the IRF model is positive under RCP8.5 and also partly positive under the other two scenarios. Please make the color scheme in Fig. 5 consistent with the attribution as described in the main text and in Tab. 1 and with the text on line 389 to 392.

Yes, we agree that section 2.4 and Figure 5 were not consistent. Instead of changing the figure, we have decided that it is clearer to revise section 2.5 to clarify that we can define $C_{transport}$ only when the sum of the CESM emulation ($C_{total}$) and the warming and chemistry effects remain less than the historical scaling. Otherwise, the upper bound on potential uptake by the ocean is simply the no chemistry change, no warming run ($C_{ccc}$). We have made other minor edits throughout the text to integrate this improved presentation.

4) "..transport to begin to play a role after 2060" suggest to rewrite to "..transport effect to begin to play a role after 2060" and similar on line 561

Thank you, we have made this change on both lines.

Thank you again for your careful attention to this manuscript.

Sincerely,

Galen McKinley, Professor, mckinley@ldeo.columbia.edu
Sean Ridge

P.O. Box 1000      61 Route 9W      Palisades, NY 10964-8000 USA      845-329-2900      http://www.ldeo.columbia.edu